Corrected: Author correction

# Structural determinants and functional consequences of protein affinity for membrane rafts

Joseph H. Lorent[1], Blanca Diaz-Rohrer[1], Xubo Lin [1], Kevin Spring[1], Alemayehu A. Gorfe[1], Kandice R. Levental[1] & Ilya Levental [1]

Eukaryotic plasma membranes are compartmentalized into functional lateral domains, including lipid-driven membrane rafts. Rafts are involved in most plasma membrane functions by selective recruitment and retention of specific proteins. However, the structural determinants of transmembrane protein partitioning to raft domains are not fully understood. Hypothesizing that protein transmembrane domains (TMDs) determine raft association, here we directly quantify raft affinity for dozens of TMDs. We identify three physical features that independently affect raft partitioning, namely TMD surface area, length, and palmitoylation. We rationalize these findings into a mechanistic, physical model that predicts raft affinity from the protein sequence. Application of these concepts to the human proteome reveals that plasma membrane proteins have higher raft affinity than those of intracellular membranes, consistent with raft-mediated plasma membrane sorting. Overall, our experimental observations and physical model establish general rules for raft partitioning of TMDs and support the central role of rafts in membrane traffic.

[1] McGovern Medical School, University of Texas Health Science Center, Houston MSB 4.202A, 6431 Fannin St, Houston, TX 77096, USA. Correspondence and requests for materials should be addressed to I.L. (email: ilya.levental@uth.tmc.edu)

Membrane rafts are lipid-driven membrane domains that are involved in nearly all aspects of mammalian membrane physiology[1]. These domains result from preferential interactions between saturated lipids, sterols, and glycosylated lipids, while the interactions between these lipids and unsaturated phospholipids are relatively disfavored. Although pairwise interactions are relatively weak[2, 3], their collective effect results in formation of mesoscopic domains in both biomimetic[4] and biological[5, 6] membranes. Recent breakthroughs in spectroscopic imaging[7–9], single-molecule tracking[10, 11], super-resolution microscopy[12] and spectroscopy[13], lipidomics[14, 15], electron microscopy[16], in silico modeling[17], and imaging of subcellular organelles[18] have provided strong evidence supporting raft existence and physiological relevance[19]. Despite this accumulating evidence, it should be emphasized that the precise nature and functions of raft domains in living cells remain controversial, with some findings contradicting the hypothesis of cholesterol-rich domains on the cell surface[20, 21].

Part of the reason for the continuing controversy surrounding membrane rafts is the ambiguous and non-quantitative methodology used to probe their composition. Specifically, a major question remains: which proteins partition to lipid rafts and why? This question is of fundamental importance because the functionality of rafts inherently depends on their selective recruitment of proteins into membrane sub-compartments of distinct composition. Previous estimates suggest that the majority of transmembrane proteins are excluded from raft domains[22, 23] as a consequence of the tighter lipid packing therein, suggesting that specific protein features are required for raft affinity. Some features—namely palmitoylation[22] and transmembrane length[24]—have been described, but there remain few general insights about the structural determinants of raft partitioning[25].

A recent conceptual and methodological advance for the raft field is the observation of coexisting liquid-ordered ($L_o$) and liquid-disordered ($L_d$) phases in intact plasma membranes known as Giant Plasma Membrane Vesicles (GPMVs). The relatively ordered[26, 27], less diffusive[28] $L_o$ phase in these vesicles is enriched in predicted raft lipids and proteins[5, 11, 22, 29], and has therefore been termed the 'raft phase'. Conceptually, the observation of liquid-ordered domains in membranes of biological complexity and protein content confirms a central principle of the lipid raft hypothesis. Methodologically, this model system enables measurements of component partitioning between raft and non-raft domains in a near-native membrane environment. It should be noted that GPMVs do not faithfully represent all features of the intact cell plasma membrane[30], in that they lose strict leaflet asymmetry, they are at chemical equilibrium, and they lack a densely associated actin cytoskeleton network[31]. Interestingly, there is accumulating evidence that suggests this membrane-associated cytoskeleton may be one reason that live cell PMs do not separate into microscopic domains as GPMVs do. Experimental[32–34] and theoretical[35, 36] studies suggest that proteins and

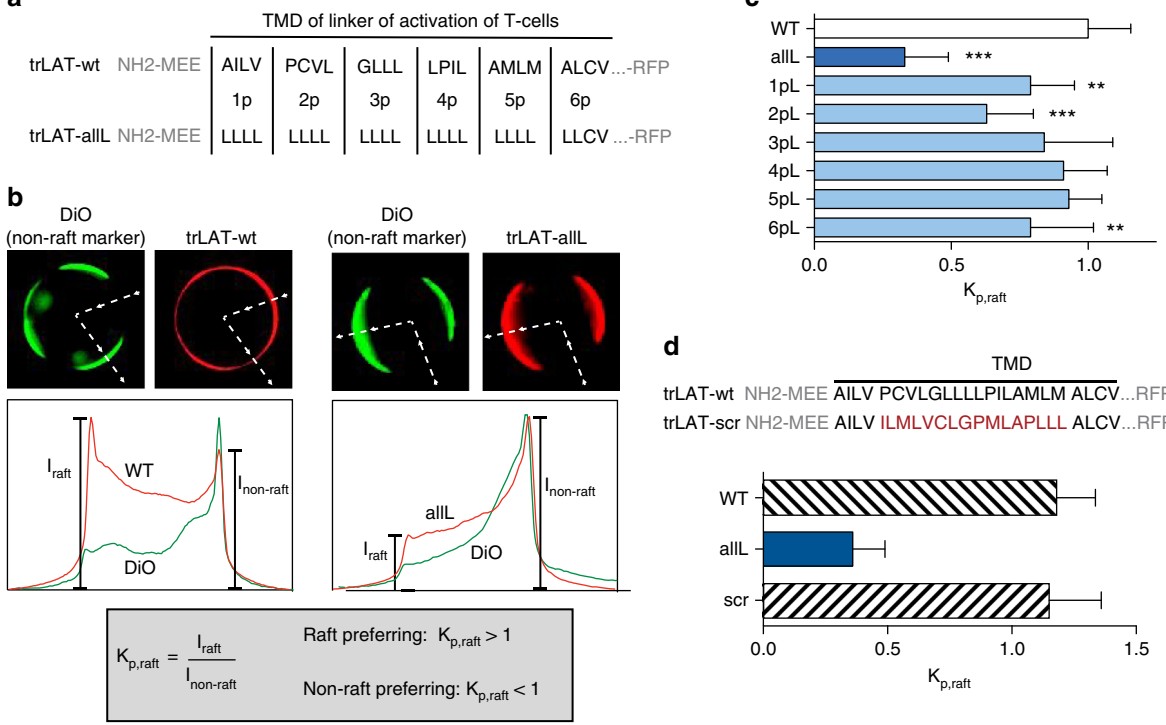

**Fig. 1** Raft-targeting features distributed over the TMD of LAT. **a** Sequences of TMDs used to determine the localization of the raft-targeting feature. Mutations were made to either the entire TMD sequence (trLAT-allL) or to individual amino acid quartets (e.g. 1pL). **b** Example of quantification of raft partitioning in GPMVs. A protein of interest (trLAT; red) is expressed in cells, which are then stained with a lipid dye (F-DiO; green) with known phase preference (non-raft phase for F-DiO). The dye is used to identify the non-raft phase in GPMVs, and the relative fluorescence intensity of the protein in the raft versus non-raft phase gives the raft partition coefficient, $K_{p,raft}$. Mutating all TMD residues to Leu (trLAT-allL) decreases the raft affinity relative to the native LAT TMD (trLAT-wt). Vesicles in images are 5–10 μm in diameter. **c** The TMD was divided into 6 parts, that were mutated individually to Leu to identify the location of the raft-targeting features. None of these partial mutations reproduced the lack of raft affinity of the all-Leu construct, suggesting a distributed feature responsible for the raft affinity of the LAT TMD. **d** The 16 core residues of the LAT TMD were randomized to create trLAT-scr. This construct partitioned at parity with the wild-type TMD, suggesting amino acid properties, rather than sequence, as the key determinant of raft affinity. Average±SD for 3–5 independent trials, each with > 10 vesicles/condition; **$p<0.01$; ***$p<0.001$, one-way ANOVA. Sequences and partitioning values for all variants are given in Supplementary Table 1

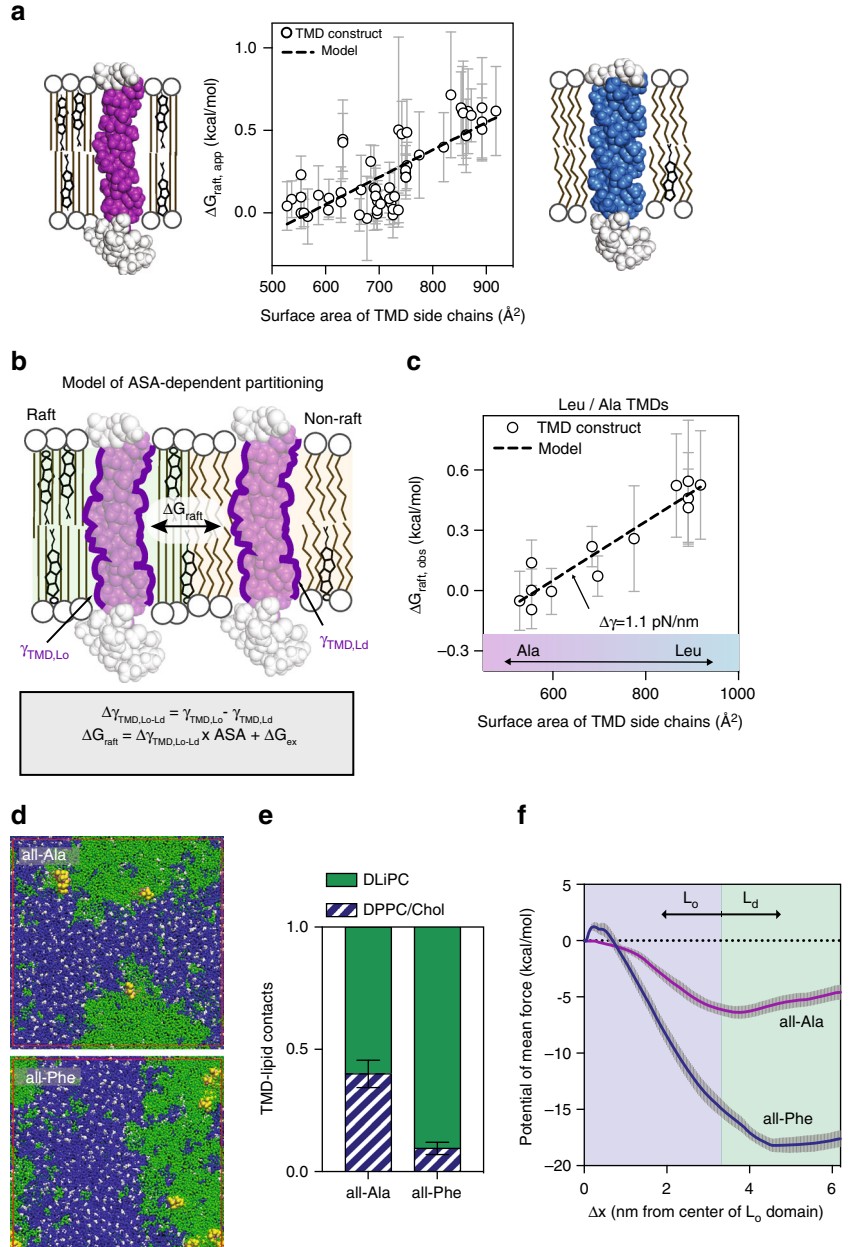

**Fig. 2** TMD surface area determines raft partitioning. **a** For 56 TMD constructs, computed side chain surface area correlates strongly with raft affinity ($r = 0.74$, $p < 0.001$), with TMDs with smaller areas preferentially partitioning to the raft phase. **b** Schematic model for accessible surface area (ASA)-dependent raft affinity based on differential interfacial tension ($\Delta\gamma_{TMD,Lo-Ld}$) between a TMD and the membrane matrix in raft versus non-raft phases. **c** Apparent raft affinity $\Delta G_{raft,app}$ for 12 constructs whose TMDs are comprised solely of Ala and Leu residues shows a clear dependence on TMD side chain surface area. The dotted line shows the model fit, which yields a prediction for $\Delta\gamma_{TMD,Lo-Ld}$ of 1.1 pN/nm ($r = 0.93$, $p<0.001$). Average±SD for > 10 vesicles/condition representative of 3–5 independent trials. Sequences and partitioning values for all variants are given in Supplementary Table 2. **d** Endpoint snapshots of CG MD simulations of model TMDs (yellow) with small (all-Ala) and large (all-Phe) surface areas in a phase separated membrane composed of 50% DPPC (blue), 30% DLiPC (green), and 20% cholesterol (white). **e** Quantification of contacts between TMDs and lipids reveals that all-Ala interacts more avidly with $L_o$ lipids (DPPC+Chol) than $L_d$ lipids (DLiPC). **f** Umbrella sampling calculation of relative free energy (PMF) difference resulting from translating a TMD from the center of the $L_o$ domain ($x = 0$ nm) to the $L_d$ ($x = 6$ nm). all-Ala has much higher relative $L_o$ affinity than all-Phe

lipids immobilized by their association with the cytoskeleton may prevent large-scale phase separation, instead restricting lipid-driven domains to sub-microscopic length scales in live cells. One inference from this hypothesis is that the microscopic raft phases in GPMVs are a reasonable model for the dynamic, nanoscopic domains that are important for the organization and functionality of live cell PMs. This possibility has been supported by several recent elegant studies. Two independent labs showed that

fluorescent lipid analogs that enrich in $L_o$ domains in GPMVs are also detergent-resistant (the biochemical hallmark of raft affinity), and most importantly show diffusive behaviors in live cells corresponding to nanoscopic domains[10, 11]. These findings strongly suggest the presence of nanoscale domains in live cells, which are manifested as coalesced $L_o$ phases in GPMVs. These domains, and their recruitment of specific proteins, were shown to be functionally important for viral binding and fusion[37]. Another set

of observations showed that proteins which partition to the ordered-phase in GPMVs form raft-like clusters in intact cells and that these behaviors are associated with immune cell function[12].

Finally, a combination of detailed lipidomic and physical investigations of isolated GPMVs showed that PM phenotypes are distinct even in closely related cell lineages, and can be used to affect lineage specification in mesenchymal stem cells[38]. Thus, GPMVs provide a tractable and coherent model system for assaying the lipid and protein composition of raft domains in a biological membrane.

To explore the structural determinants and biological consequences of raft affinity, we used this model system to probe raft phase partitioning of dozens of single-pass transmembrane protein variants. We identified novel raft-targeting features that were used to construct a predictive physical model for raft affinity. Application of this model to the human proteome suggests that raft association is related to sub-cellular sorting of transmembrane proteins, supporting a functional role of raft domains in membrane traffic.

## Results

**Identification of distributed raft-targeting feature in TMDs.**
We hypothesized that as TMDs are the predominant membrane-interacting region of a protein, this domain should play an important role in raft affinity. To investigate this hypothesis, we measured raft partitioning for transmembrane protein constructs based on the TMD of the Linker for Activation of T-cells fused to mRFP (trLAT)[22, 24] (Fig. 1a). Rat basophilic leukemia (RBL) cells were transfected with TMD constructs (Supplementary Table 1) and stained with a non-raft marker (FAST-DiO; green), followed by isolation of GPMVs, which showed the expected large-scale separation into coexisting raft and non-raft domains (Fig. 1b; green). Quantification of relative protein concentration between these two domains (via RFP intensity) provides a direct measure of the equilibrium raft partition coefficient ($K_{p,raft}$)[5]. As expected[22, 24], a construct comprised of the LAT TMD (trLAT-wt) partitioned slightly preferentially to the raft domain. In striking contrast, a construct whose TMD consisted solely of leucine residues (trLAT-allL) showed greatly reduced raft affinity (Fig. 1b), confirming that the LAT TMD contains a raft-targeting feature. To localize this feature, we mutated the LAT TMD to Leu within six separate amino acid quartets (Fig. 1a). Surprisingly, none of these partial mutations resulted in wholesale abrogation of raft affinity observed for the allL TMD (Fig. 1c), suggesting that the raft-targeting feature was distributed over the TMD. Thus, we hypothesized that domain partitioning may be related to physical features of the TMD as a whole. To test this possibility, we constructed a "scrambled" TMD that maintained the residue identity of the wild-type LAT TMD, but randomized the sequence of the 16 amino acids embedded in the hydrophobic core of the membrane (Fig. 1d). Remarkably, this construct partitioned at parity with the wild-type, suggesting that the aggregate physicochemical features of the amino acids, rather than their specific sequence, was required.

**TMD surface area is a determinant of raft affinity.** We noted that while partial Leu mutations of the TMD slightly disrupted raft partitioning, previous reports of Ala mutation of similar residues[24] had no such effect (Supplementary Fig. 1). Leu bears a much larger aliphatic side chain than Ala, suggesting that the physical property governing domain preference may be related to the steric size of TMD amino acids. To evaluate this hypothesis, we measured the apparent free energy of raft partitioning ($\Delta G_{raft,app} = -RT \ln K_{p,raft}$) for a variety of TMDs (Fig. 2a) and observed

a remarkable correlation ($p<0.0001$) between raft phase affinity and total solvent accessible surface area (ASA) of the TMD side chains (calculated from bioinformatic/computational predictions[39]) and validated by molecular modeling (Supplementary Fig. 2). Specifically, proteins with smaller TMD surface area showed greater raft affinity than larger ones (Fig. 2a).

We rationalized these observations with a simple mechanistic model wherein the energetics of partitioning between coexisting membrane domains ($\Delta G_{raft}$) are driven by the differential interfacial tension ($\Delta\gamma_{TMD,Lo-Ld}$) between a TMD peptide solvated by a raft ($\gamma_{TMD,Lo}$) versus non-raft ($\gamma_{TMD,Ld}$) membrane environment (Fig. 2b),

$$\Delta G_{raft} = \Delta\gamma_{TMD,Lo-Ld} \times ASA_{TMD} + \Delta G_{ex} \qquad (1)$$

where $\Delta G_{ex}$ is a constant whose physical origin is described below. We observe very good agreement between this model (dotted line in Fig. 2a) and the experimentally observed partitioning of LAT-derived TMDs. We note that despite the generally strong agreement between data and model, there is a cluster of constructs around 700 Å$^2$ whose raft affinity is somewhat higher than predicted by the model. All of these LAT-based constructs contain Pro and Gly residues, which may have subtle effects on TMD conformation and thus slightly affect partitioning.

To control for this possibility and corroborate the ASA model, we designed 12 constructs with TMDs comprised solely of Leu and Ala residues whose relative abundance was tuned to yield varying surface areas. We observed a robust and significant dependence ($R^2 = 0.87$, $p<0.0001$) of raft affinity on TMD ASA for these simple constructs (Fig. 2c) that was well described by the model in Fig. 2b (dotted line). Remarkably, a TMD comprised solely of 14 Ala and 8 Leu (allA+8L) reproduced the raft affinity of wild-type LAT (Supplementary Table 2). The slope of the model fit yields an experimental estimate of $\Delta\gamma_{TMD,Lo-Ld}$—the difference in interfacial tension between TMDs dissolved in a raft versus non-raft environment—of 1.1 pN/nm. This estimate is in good agreement with a previous computational prediction of interfacial tension between $L_o$ and $L_d$ domains (0.15 kT/nm$^2$; 0.6 pN/nm)[40].

We also tested a potential correlation with the surface roughness of the TMDs, since roughness can be strongly related to surface area depending on the length scales being probed. TMD roughness was measured via the fractal dimension method[41], which allows calculation of roughness over a variety of length scales. We measured roughness between 5 and 6 Å, reasoning that the relevant length scale for lipid–TMD interactions would be the persistence length of a lipid in a bilayer, previously estimated to be ~ 5 Å[42]. At this length scale, TMDs are nearly smooth (fractal dimension = 2 is a featureless surface) and there is a much weaker dependence of $\Delta G_{raft}$ on roughness (Supplementary Fig. 3) compared with ASA.

Because nearly all TMD residues are hydrophobic, and larger residues with higher ASA are also more hydrophobic, there is an inherent, strong correlation between TMD ASA and overall TMD hydrophobicity. Consistently, TMD hydrophobicity (calculated via the scale of Kyte and Doolittle[43]) was also strongly correlated with raft affinity for the Ala/Leu constructs described above and in Fig. 2c (black circles in Supplementary Fig. 4b). To determine whether TMD ASA or hydrophobicity was the major driver of raft affinity, we introduced large charged or hydrophilic residues into TMDs. Specifically, we added two Lys residues into the allL construct (allL2K) and 4 Trp residues into the allA8L construct (allA8L4W). In both cases, the observed raft partitioning was much closer to predictions from the ASA of the TMD rather than the hydrophobicity (Supplementary Fig. 4). These experiments

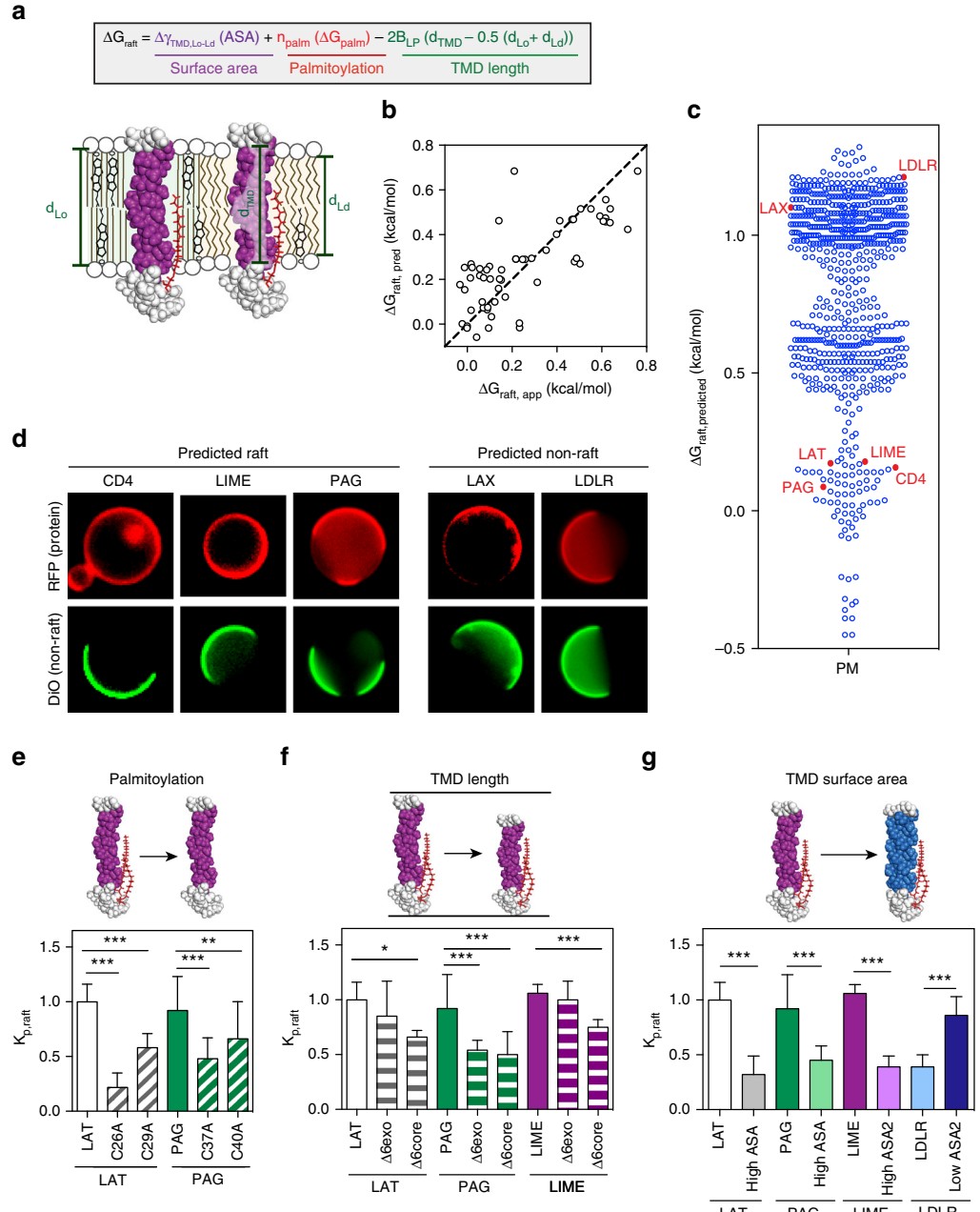

**Fig. 3** Construction of a comprehensive, predictive raft partitioning model. **a** TMD length and palmitoylation have been previously identified as determinants of raft partitioning. These have been formalized together with the surface area model from Fig. 2 into a comprehensive description of raft partitioning for TMDs. **b** Using parameters derived from published data and Fig. 2, raft affinity predicted from the tripartite model ($\Delta G_{raft,pred}$) shows excellent agreement with measured values ($\Delta G_{raft,app}$) for all tested TMDs (i.e., the constructs from Fig. 2a plus 4 palmitoylation mutants and 4 TMD truncations) without any floating fit parameters. **c** The model was used to predict raft affinity for the 729 single-pass PM proteins in the human proteome, from which three were chosen as representative raft-preferring proteins and two as non-raft preferring. **d** The partitioning of constructs containing TMDs from these five proteins followed predicted trends, with all three raft-predicted TMDs (CD4, LIME, PAG) partitioning efficiently to the raft domains, while both predicted non-raft proteins (LAX and LDLR) showed minimal raft partitioning. Vesicles in images are 5–10 μm in diameter. **e–g** The characteristics determining the raft partitioning of these constructs were dissected by mutating the TMDs to modify palmitoylation, TMD length, and ASA. **e** Mutating either of two palmitoylation sites reduced raft partitioning for LAT and PAG; **f** truncating the TMD length by 6 amino acids reduced raft partitioning for LAT, PAG, and LIME (but not LIME-Δ6exo); and **g** increasing ASA reduced raft partitioning for PAG and LIME, while decreasing ASA significantly increased raft partitioning for LDLR. Average±SD for 3–5 independent trials, each with > 10 vesicles/condition; *$p<0.05$; **$p<0.01$; ***$p<0.001$, one-way ANOVA. Sequences and partitioning values for all variants are given in Supplementary Tables 3, 4

provide strong evidence that TMD surface area is the dominant parameter determining TMD raft affinity.

Although these experiments clearly implicate TMD surface area as a determinant of ordered domain partitioning, these observations are made in the complex milieu of an isolated plasma membrane, and without any direct information about the structure of the assayed proteins. Therefore, we directly verified the effect of TMD surface area on membrane domain partitioning by computational experiments on a strictly defined system. Specifically, we used coarse-grained molecular dynamics

simulations to investigate the interactions between TMDs and a phase-separating membrane consisting of saturated lipids (dipalmitoylphosphatidylcholine; DPPC; 50%), unsaturated lipids (dilinoleylphosphatidylcholine; DLiPC; 30%), and cholesterol (20%). We probed two model TMDs, representing extremes of small (all-Ala) and large (all-Phe) ASAs. Consistent with experiments[4] and previous simulations[40, 44], this mixture quickly (within 1 μsec) separated from an initially homogeneous state into stable $L_o$ and $L_d$ domains enriched in DPPC/cholesterol and DLiPC, respectively (Fig. 2d). We quantified the relative distribution of the TMD peptides between these domains by the number of molecular contacts between the peptides and $L_o$ (DPPC/Chol) versus $L_d$ (DLiPC) phase lipids. This analysis revealed that while the all-Phe TMD showed negligible interaction with $L_o$ lipids, the all-Ala TMD interacted approximately equally with both phases (Fig. 2e). We complemented these observations by using umbrella sampling to calculate the potential of mean force (PMF) associated with translating a TMD from the $L_o$ to the $L_d$ phase, yielding estimates for relative free energy of ordered phase partitioning. While both all-Ala and all-Phe had a smaller PMF (i.e., lower energy) in the $L_d$ domain, the smaller all-Ala had much higher relative $L_o$ affinity than all-Phe (Fig. 2f).

**A comprehensive model for TMD raft partitioning.** The model of surface area-dependent partitioning implies that there is an inherent energetic cost to transferring a TMD from a non-raft to a raft domain, consistent with the tighter lipid packing therein. However, our experimental measurements show that some TMDs partition robustly to the raft phase, suggesting a mechanism for enhancing raft affinity, represented in the model by the parameter $\Delta G_{ex}$. We propose two distinct physical bases for this effect: acylation by saturated fatty acids and hydrophobic matching between TMD length and membrane thickness.

A number of previous reports have implicated post-translational modification of TM proteins by saturated fatty acids (i.e., palmitoylation) as a key driver of raft association[45]. The putative physical basis for this effect is that saturated fatty acids interact favorably with cholesterol[2, 46, 47] and therefore increase partitioning of palmitoylated proteins to the cholesterol-rich raft phase. This effect has been directly demonstrated in raft partitioning experiments on isolated plasma membranes[22], from which we derive an experimental estimate for the raft partitioning free energy due to palmitoylation ($\Delta G_{palm}$) of −0.48 kcal/mol per acylation (calculation details in legend of Supplementary Fig. 5). We have previously investigated whether TMD mutations affect palmitoylation and found that all TMD constructs had similar palmitoylation levels[24], suggesting that TMD-dependent effects on raft association are independent of changes in palmitoylation.

Similarly, it has been recently shown that the length of a protein TMD is a determinant of raft partitioning, with longer TMDs preferring more ordered domains[24, 48]. This effect can be rationalized by invoking the hydrophobic mismatch principle, through the application of a simplified version of the "mattress" model[49], which postulates an energetic cost for incompatibility between the length of a TMD and the hydrophobic thickness of the surrounding membrane. This mismatch is minimized in a membrane with domains of different thicknesses—as in the $L_o/L_d$ model membranes[50–52] and presumably also in GPMVs—by longer TMDs partitioning to thicker domains. This simple model fits well to published experimental data[24] and was used to derive the parameters for TMD length-based partitioning (Supplementary Fig. 5).

The three independent structural determinants (TMD length, surface area, and palmitoylation) were combined into the comprehensive model for raft partitioning of single-pass

transmembrane proteins shown in Fig. 3a:

$$\Delta G_{raft,pred} = \Delta\gamma_{TMD,Lo-Ld}(ASA_{TMD}) + n_{palm}(\Delta G_{palm}) \\ -2B_{LP}(d_{TMD} - 0.5(d_{Lo} + d_{Ld})) \quad (2)$$

where $n_{palm}$ is the number of palmitoylated cysteines, $B_{LP}$ is the hydrophobic mismatch parameter from the mattress model[49], and $d_{TMD}$, $d_{Lo}$, and $d_{Ld}$ are the lengths of the TMD, $L_o$, and $L_d$ phases, respectively (for details on the calculations of these parameters, see Supplementary Fig. 5 and associated legend).

Applying this model to the panel of ~ 60 TMD constructs measured here and in previous reports[22, 24] yields very good agreement between predicted and experimentally observed partitioning (Fig. 3b). Significantly, this agreement was obtained without any additional fit parameters; the length and palmitoylation parameters were obtained from published data, while Δγ was obtained from the fit in Fig. 2c.

To test whether this model would be applicable to unrelated TMDs, we generated predictions for raft partitioning for all predicted single-pass PM proteins in the human proteome (Fig. 3c). The distribution showed a major cluster of proteins with relatively low raft affinity ($\Delta G_{raft,pred} > 1$ kcal/mol), and at least two smaller populations with higher raft affinity. This prediction that the large majority of PM proteins are not raft associated is consistent with previous estimates[22, 23]. From the evaluated set of PM proteins, we chose two TMDs predicted to have low raft affinity (LAX and LDLR) and three (PAG, LIME, and CD4) whose predicted raft affinity was similar to that of LAT ($\Delta G_{raft,pred}$ ~0.1 kcal/mol). The partitioning of these TMDs was evaluated, and for all five, the observed partitioning in GPMVs qualitatively matched the prediction of the model (Fig. 3d). This validation confirms that our tripartite model successfully predicts raft partitioning trends for single-pass PM proteins solely from primary amino acid sequence. Such bitopic proteins comprise ~ 50% of all transmembrane proteins[53], which in turn comprise ~ 30% of the proteome[45], suggesting that our model could have broad utility for identifying raft associated proteins. It is important to point out that because our analysis was restricted to single-spanning membrane proteins, we can make no statement about the generality of our model to multiple-span proteins.

To individually validate all three aspects of the model, we created variants of LDLR, PAG, and LIME in which one of the three parameters was modified. For all mutants tested, we observed the predicted effects. Mutation of palmitoylation sites dramatically reduced raft affinity for LAT and PAG (Fig. 3e). Truncation of TMDs by 6 amino acids significantly reduced raft partitioning for both PAG and LIME, to a similar extent as was observed for LAT (Fig. 3f). Finally, increasing TMD surface area for both PAG and LIME by Ala-to-Leu substitutions essentially abrogated raft partitioning. Most remarkable, decreasing TMD surface area of the non-raft TMD of LDLR (by Leu-to-Ala mutations) significantly increased its raft partitioning to near-parity with wild-type LAT TMD (Fig. 3g). Thus, rational re-engineering of protein TMDs was able to create a raft-preferring protein from a non-raft scaffold.

**Proteome-wide prediction of raft affinity.** Having confirmed our predictive model for raft affinity, we applied it to all single-pass TM human proteins to evaluate the long-standing association between membrane domains and subcellular membrane traffic. The raft hypothesis was originally formulated as a mechanistic explanation for the unique lipid and protein composition of the apical plasma membrane in epithelial cells[54], and was later expanded to encompass PM traffic in non-polarized

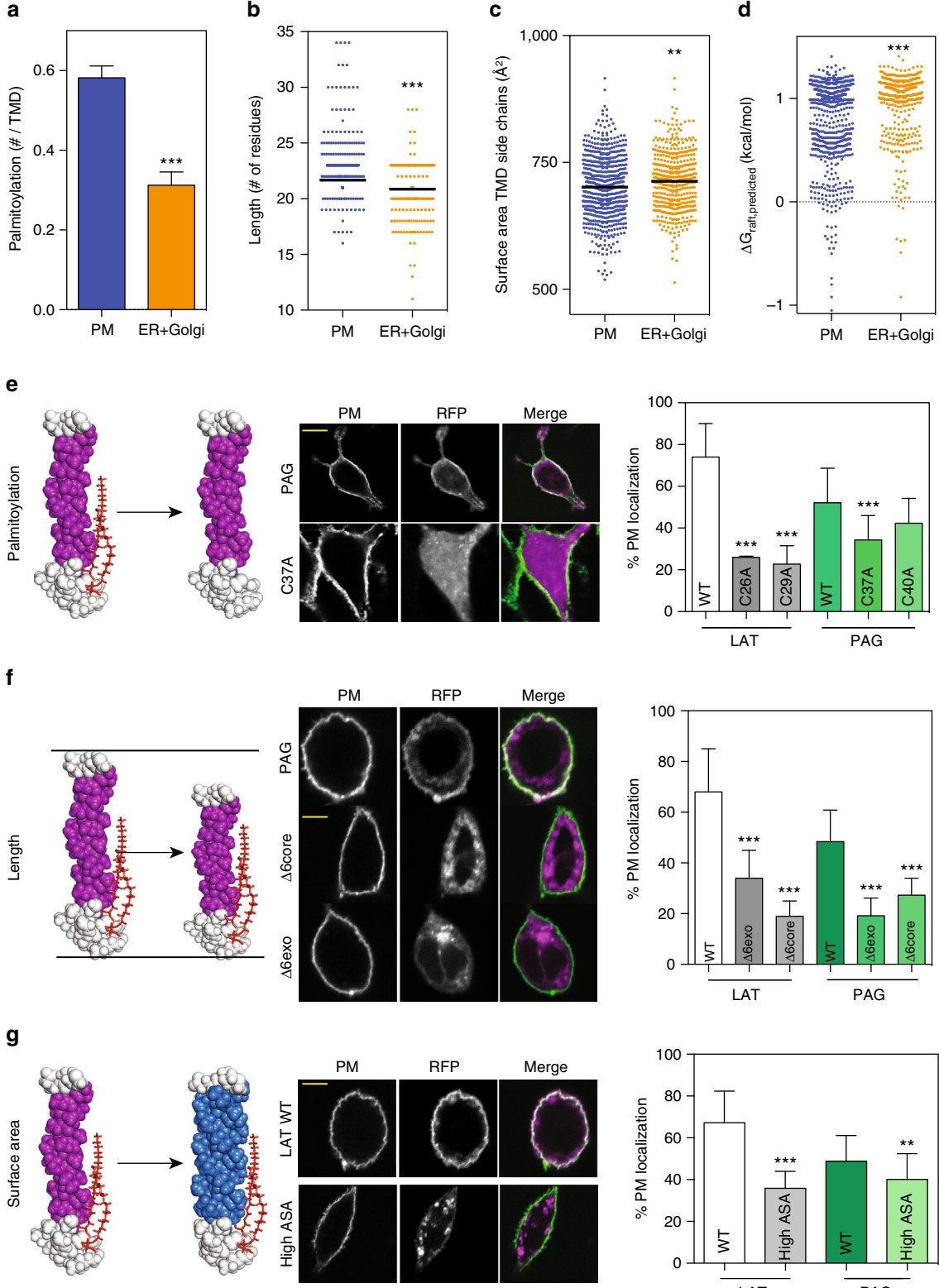

**Fig. 4** PM proteins have higher raft partitioning, and mutations of raft affinity reduced PM localization. Proteome-wide bioinformatic predictions of **a** palmitoylation, **b** TMD length, and **c** TMD side chain surface area reveal that all features associated with raft partitioning are predicted to be significantly enhanced in single-pass PM proteins versus a combined set of single-pass ER and Golgi proteins. **d** Consistently, predicted raft affinity is significantly higher for PM proteins than non-raft (3 outliers omitted for clarity). Significances are one-way ANOVA relative to PM; ***$p<0.001$; **$p<0.01$. The distribution of $\Delta G_{raft}$ is clearly not unimodal; histogram with Gaussian fits in Supplementary Fig. 6. **e–g** Experimental evaluation of bioinformatics predictions. PM localization of two predicted raft proteins (LAT and PAG) was significantly reduced by mutating any of the structural features determining raft partitioning, including **e** palmitoylation, **f** TMD length, and **g** TMD surface area. Scale bars are 5 µm. Avg ± SD for 20–40 cells/condition. Significances are one-way ANOVA relative to the wild-type TMD; ***$p<0.001$; **$p<0.01$

cells[14, 55, 56]. In this formulation, membrane domains serve as sorting platforms to laterally organize lipids and membrane proteins in order to facilitate their coordinated delivery to a specific cellular location. The PM is enriched in many putative raft components (incl. cholesterol, sphingolipids, and GPI-anchored proteins); therefore, it has been hypothesized to be the destination for raft-mediated membrane traffic[57]. To evaluate this hypothesis, we compared the structural features associated with raft affinity between PM proteins and those of major intracellular organelles (endoplasmic reticulum and Golgi apparatus). All three raft-associated features were over-represented in PM proteins: PM proteins were significantly more palmitoylated (Fig. 4a), and their TMDs were significantly longer (Fig. 4b) with smaller surface areas (Fig. 4c) than those of internal organelles. These observations are fully consistent with previous analyses of organelle-specific differences in TMD length and sequence[58, 59]. Combined, the differences in TMD area, length, and palmitoylation led to dramatically different distributions of predicted raft affinities, with PMs proteins having significantly higher average raft affinity (i.e., lower $\Delta G_{raft,pred}$) (Fig. 4d). This effect was not due to an overall shift in the distribution of raft affinities, as the large non-raft population was similar between the two groups, but rather a higher abundance of predicted raft-preferring proteins in PMs (Supplementary Fig. 6).

## Discussion

It is a remarkable observation that the very same protein features that determine raft affinity (longer TMD length, smaller TMD surface area) were previously identified as determinants of PM localization in a landmark biostatical study by Sharpe et al.[58]. This work identified a fundamental cellular design principle, observed in both fungi and mammals, that certain organelles possess distinct protein TMD features. We believe our results support a mechanistic explanation underlying this principle, specifically that PM localization for certain proteins is driven by raft-mediated sub-cellular sorting/trafficking. This inference and the biostatistical prediction in Fig. 4d is supported by direct experimental evidence relating subcellular localization with raft affinity. Specifically, we quantified plasma membrane localization of TMD constructs by expressing RFP-tagged TMDs in RBL cells and imaging by confocal microscopy. The plasma membrane was stained by surface biotinylation and fluorescent streptavidin (Fig. 4e–g) and the abundance of each TMD at the PM relative to intracellular membranes was quantified by an automated image processing protocol. Both TMDs chosen for this analysis (LAT and PAG) were from expected PM proteins and the constructs indeed largely localized to the PM. In contrast, abrogation of raft affinity by mutation of palmitoylation sites (Fig. 4e), decreasing TMD length (Fig. 4f), or increasing TMD surface area (Fig. 4g) significantly reduced PM residence.

These effects are fully consistent with the hypothesis that trafficking to the PM is at least partially mediated by raft domains, potentially explaining the statistically observed enrichment of long, thin TMD-containing proteins in the PM[58]. It is useful to point out that the protein constructs employed here contain only the membrane-embedded TMDs and six intracellular amino acids, and yet retain the PM localization of the native full-length proteins. Further, we have previously shown that a "minimal" construct containing only the TM residues of native LAT partitions to rafts and localizes at the PM similarly to the full length protein[24]. These data lead to two important conclusions about this set of proteins. First, their TMDs are fully sufficient for raft partitioning (though it is likely that extramembrane protein regions are important for raft partitioning in other contexts, e.g., if they mediate tight protein–protein interactions). Second, as no extramembrane residues are necessary for PM localization, these proteins are not sorted by coat/adapter machinery and thus likely traffic to the PM via membrane rafts.

In conclusion, our direct experimental measurements of raft partitioning in isolated plasma membrane vesicles reveal several independent structural features associated with raft affinity for single-pass transmembrane proteins. Namely, we find that generalized physicochemical features of protein TMDs—palmitoylation, length, and surface area—are determinants of raft affinity. These features were reconstructed into a model that was effective in predicting raft affinity from primary amino acid sequence. This model applied to the human proteome reveals enrichment of raft preferring proteins at the PM, and these predictions were confirmed by experimental quantification of trafficking defects induced by mutation of raft-targeting features. These observations establish the general principles that govern raft affinity, generate predictive estimates for raft residence, suggest a mechanistic explanation for a previously observed cellular design principle, and validate the central role of raft-associated trafficking in subcellular sorting of membrane proteins.

## Methods

**Cell culture and transfection.** Rat basophilic leukemia (RBL) cells were purchased from ATCC and cultured in medium containing 60% Eagle's Minimum Essential Medium (MEM), 30% RPMI, 10% FCS, 100 U/ml penicillin, and 100 µg/ml streptomycin at 37 °C in humidified 5% $CO_2$. Transfection was done by nucleofection (Amaxa) using the protocols provided with the reagents. After 4–6 h of transfection, cells were washed with PBS and then incubated with serum-free medium overnight. To synchronize the cells, 1 h before preparation of GPMV, the cells were given full-serum medium.

**Partitioning measurements in Giant Plasma Membrane Vesicle.** Cell membranes were stained with 5 µg/ml of FAST-DiO (Invitrogen), a green fluorescent lipid dye that strongly partitions to disordered phases[60]. Following staining, GPMVs were isolated from transfected RBLs as described[5, 6]. Briefly, GPMV formation was induced by 25 mM formaldehyde/2 mM DTT in isotonic buffer containing NaCl, 10 mM HEPES, and 2 mM $CaCl_2$, pH 7.4. To quantify protein partitioning, GPMVs were observed on an inverted epifluorescence microscope (Nikon) at 10 °C. The partition coefficient ($K_{p,raft}$) for each protein construct was calculated from fluorescence intensity of the RFP construct in the raft ($I_{Lo}$) and non-raft ($I_{Ld}$) phase for >10 vesicles/trial (Fig. 1). At least two independent experiments were performed for each construct and the values were normalized to WT trLAT in each experiment. Raft phase affinity was calculated as $\Delta G_{raft,app} = -RT \ln K_{p,raft}$.

Expression level/TMD concentration did not have an effect on partitioning, as evidenced by the lack of correlation between RFP fluorescence intensity in GPMVs (as a proxy for the abundance of each TMD construct) and partitioning (Supplementary Fig. 7). The lack of dependence of protein partitioning on concentration despite >10-fold differences in protein concentration is strong evidence that dimerization/oligomerization has no effect on our observations, as oligomerization would be strongly affected by concentration. This is likely because none of the constructs tested here show any significant dimerization, consistent with the absence of literature reporting autonomous LAT oligomerization.

**TMD constructs.** All TMD constructs were based on the trLAT backbone previously described[22, 24]. The amino acid sequence of WT trLAT is $NH_2$-MEEAILVPCVLGLLLLPILAMLMALCVHCHRLPGS followed by a short linker (GSGS) and monomeric RFP (mRFP). TMD mutants were generated by synthesizing the gene of interest (Genscript) and subsequent cloning of the mutant sequence into the trLAT construct. Mutants were confirmed by sequencing. The sequences and partitioning values of all mutants used in this study are given in Supplementary Tables 1–4.

**Determination of parameters for raft partitioning model.** To determine which amino acids constituted the transmembrane domain, we used the annotations from the Uniprot database and predictions from TMHMM[61]. The aggregate accessible surface area of TMD side chains was calculated by summing values for individual residues, described in Yuan et al.[39]. The number of palmitoylations ($n_{palm}$) was taken as the number of Cys residues in the cytoplasmic end of the TMD–LAT has two such Cys, which are both known to be palmitoylated[62]. Previous measurements show that mutation of non-Cys TMD residues does not affect palmitoylation of LAT[24].

**Bioinformatics**. The sequences of all human single-pass transmembrane proteins localized in the plasma membrane, the Golgi apparatus and the endoplasmic reticulum were downloaded from the Uniprot database. The transmembrane domains were extracted and the sequences were aligned from exoplasmic to cytoplasmic. To determine palmitoylation, a published algorithm (CSS-palm 2.0)[63] was applied to the 15 amino acids comprising the cytoplasmic end of the TMD and membrane proximal part of the cytoplasmic domain. Length and ASA were determined by TMHMM and predictions from Yuan et al as above. The free energy of raft partitioning was predicted by the formula shown in Fig. 3.

**Molecular dynamics simulations**. Lipids and TMDs were modeled by Martini coarse-grained (CG) force field (v. 2.1)[64, 65]. The two systems modeled both contained 1449 DPPC, 864 DLiPC, 576 Cholesterol, 66789 CG water, and 0.15 mM Na$^+$/Cl$^-$ with nine doubly-palmitoylated all-Ala or all-Phe peptides. The systems were built by first randomly placing the lipids around one TMD and keeping the system at $T$=400 K for 100 ns to mimic a randomly distributed state, which was then replicated nine-fold for an initial configuration. For all simulations, a cutoff of 1.2 nm was used for van der Waals (vdW) interactions, and the Lenard–Jones potential was smoothly shifted to zero between 0.9 nm and 1.2 nm to reduce cutoff noise. For electrostatic interactions, the columbic potential was smoothly shifted to zero from 0 to 1.2 nm (cutoff = 1.2 nm). The relative dielectric constant was 15 (default value of the force field[64]). Lipids and water/ions were coupled separately to V-rescale heat baths[66] at $T$=298 K, with a coupling constant $\tau$=1 ps. The systems were simulated at 1 bar pressure using semi-isotropic Parrinello-Rahman pressure coupling scheme[67] with a coupling constant $\tau$=5 ps and the compressibility of 3×10$^{-4}$ per bar. The neighbor list for non-bonded interactions was updated every 10 steps with the cut-off 1.4 nm. All the simulations were performed for 32 μs (effective time) with a time step of 20 fs and periodic boundary conditions using GROMACS 4.5.4[68]. Snapshots of the simulation system in this paper were all rendered by VMD[69]. Contact was defined if any two CG beads from different molecules are within 0.6 nm[70].

**Umbrella sampling and PMF**. In order to explore the free energy profile of TMD translocation between L$_o$ and L$_d$ domains, PMF was calculated using umbrella sampling simulations and the weighed histogram analysis method (WHAM)[71]. For these simulations, diarachidonoyl PC (DAPC) was used instead of DLiPC because it yielded a more stable domain boundary. The center-of-mass distance along $x$ dimension between TMD and center of the L$_o$ domain was chosen as the reaction coordinate. The initial membrane system was set up using a sandwich structure of L$_o$ domain with $x$-width ~ 6.7 nm surrounded by two L$_d$ domains, which is wide enough to avoid TMD's crossing simulation box boundary. The final box dimensions are about 27 × 16×10 nm$^3$. The range for the reaction coordinates is 6.6 nm with window spacing 0.2 nm, and thus 34 simulations with different reaction coordinates were performed for umbrella sampling. A spring constant of 1000 kJ/mol/nm was used. Each simulation was run 3.2 μs with the last 2.4 μs for the WHAM analysis. Errors are estimated based on six 400 ns blocks over the last 2.4 μs.

**Plasma membrane localization of protein constructs**. PM localization was quantified by an automated imaging protocol[24]. Briefly, transfected RBL cells were surface-labeled by a membrane impermeable biotinylation reagent (Biotin-NHS-LC) at 1 mg/ml in PBS for 20 min at 4 °C. Cells were fixed with 4% paraformaldehyde and the PM were fluorescently labeled by streptavidin-Alexa488 (10 μg/ml for 20 min). Cells were imaged on a Nikon A1R confocal microscope using a ×60 Apochromat oil immersion objective. PM localization was quantified by a custom image processing protocol (MATLAB) in which the fluorescence intensity of the protein construct that was co-localized with the plasma membrane stain was divided by the total fluorescence intensity in the whole cell.

**Data availability**. The data that support the findings of this study are available from the corresponding author upon request.

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

## Acknowledgements
We acknowledge funding from the Cancer Prevention and Research Institute of Texas (R1215), Volkswagen Foundation (Grant No. 93091), and the National Institute of General Medical Sciences, National Institutes of Health (R01GM100078, 1R01GM114282, R01GM124072). We acknowledge technical and conceptual assistance from Ed Lyman, Michael Schick, Matthias Gerl, Mark Teese, and the group of Sarah Keller. We acknowledge James Saenz, Robert Ernst, and Daniel Lingwood for thoughtful feedback on the manuscript and Helena Jambor for figure design. We thank the Extreme Science and Engineering Discovery Environment (XSEDE, project: TG-MCB150054) and the Texas Advanced Computing Center (TACC) for generous computational resources.

## Author contributions
J.H.L. and I.L. designed the study. J.H.L., I.L., B.D.-R. and K.S. performed experiments. J.H.L. designed and performed the bioinformatics calculations. J.H.L., I.L. and K.R.L. wrote the manuscript. X.L. and A.A.G. designed and performed the computational simulations.

## Additional information

**Competing interests:** The authors declare no competing financial interests.

