## [Peer Review File · Nature Communications]

Reviewers' comments:

Reviewer #1 (Remarks to the Author):

The paper by Lorent et al. describes a simple physical model applicable to transmembrane proteins based on 3 measurable parameters: number of palmitoylation cysteines, length of transmembrane domain and surface area of the side chains. Such model allows the calculation of the Gibbs free energy change upon partitioning from lipid disordered to ordered phases in the plasma membrane. More precisely, the model is derived by measuring partitioning of specifically designed polypeptides between liquid-disordered and liquid-ordered phases in vesicles artificially induced from the plasma membrane (GPMV). The model is also used to predict the partition behavior in GPMV for 729 single-pass transmembrane proteins from the human proteome. Finally, the authors show that, from a practical point a view, their model can be used to predict protein distribution among different cellular membranes (i.e. plasma membrane, ER or Golgi).

The manuscript is very well written and the conclusions are compatible with the experimental results. I have few comments regarding the experimental part and the clarity of exposition.

When K_p is measured for the different constructs, is the expression of such constructs comparable? Can K_p be a function of protein concentration?

The captions should better explain the difference (e.g. overlapping) between the data in Figs. 2A and 3B

In the third paragraph of the results, the authors state that "some TMDs clearly favor the raft phase". Nevertheless, in the results, I do not see any reported ΔG of partition clearly lower than zero (or K_p significantly larger than 1). The only exception is in Figure 1D. Regarding this figure though, why is the WT K_p larger than 1 in panel D but lower (for the same WT) in panel C? In the third paragraph of the results, the authors state that the model "yields excellent agreement" with the observed partitioning. Nevertheless, Figure 3B shows a cluster of circa 20 proteins (out of circa 60) showing an overestimation of the predicted ΔG .

Regarding the role of the exposed surface area: Could there be a role played by the roughness of the surface rather than just the amount of the exposed surface? A smooth/flat surface might have more favorable interactions with the ordered lipid environment. Maybe a role of a surface-volume ratio?

Similarly, the data in Figure 2 A show that (differently from what predicted by the model) TMDs with a surface area below 750 \AA^2 have a very low ΔG of partition, independently from the actual surface area value. Can the prediction be improved in this case by including the TMD length term?

At the end of the third paragraph, the author state that they could rationally design a "raft preferring" LDLR protein. Nevertheless, this appear once again confusing since the relative K_p is not significantly higher than 1. Also at this regard – just a small note -, choosing a protein candidate from the lower part of figure 3 C might result in more impressive results for figure 3 D. On a different note, I am concerned about the originality of the conclusions and how this study can appeal a vast amount of general readers.

First, even for scientists working in the field of cell biology, the concept (i.e. the importance in a biological context) of raft domains is not accepted in general. For each study proving existence of raft domains mentioned by the authors, there is a study arguing against the general definition and existence of raft domains (as they are usually defined, e.g. Frisz et al. J Biol Chem 2013) or their role in cell biology (e.g. Wilson et al. Biophys J 2015). Without going too much into details here, also the authors state that this is a controversial topic. The model presented is validated by (and allows to predict mostly) the outcome of experiments in which partition of proteins is determined between artificial lipid phases in membrane system whose biological relevance is at least questionable. How useful is it to predict whether a certain protein will partition in the ordered phase of a GMPV or not? If the authors had showed instead that the model allows the prediction of a generally relevant and measurable outcome (e.g. involvement of a protein in a signaling pathway), the study would have had a much higher relevance.

Second, the author do indeed provide evidence that their model allows also the prediction of more

general and interesting features, namely intra-cellular membrane distribution of proteins and their partition in the plasma membrane. This section of the results is of course much more relevant. I find though that this part is not entirely original and innovative: Intracellular sorting of membrane proteins predicted by TMD length and palmitoylation has been reported in the past (e.g. Chum et al. J of Cell Science 2015). Sharpe et al. (Cell, 2010) proposed a model that uses the primary structure of a transmembrane protein to predict whether it will be found in the (asymmetric raft-like) bilayer environment of the plasma membrane.

In conclusion, this study is undoubtedly of very high quality and it clarifies in a quantitative way some ideas that were circulating since some time (in the circle of people working with the model of raft/liquid ordered domains in the plasma membrane). I only doubt whether this study might be of general interest in cell biology.

Reviewer #2 (Remarks to the Author):

This paper links features of transmembrane helices to their propensity to partition into ordered parts of giant plasma membrane vesicles, and extends these observations to larger classes of proteins and a physical model that describes partitioning at the level of differential surface tensions. This is quite intriguing and supported by a significant amount of data. I have no major comments and it will be interesting to see the scientific community follow up on the ideas in this paper.

Reviewer #3 (Remarks to the Author):

General comments

The work by Leventhal and col. tackles in a systematic manner and with unprecedented detail the structural features that account for single transmembrane domain (TMD) proteins preference for lipid raft domains in mammalian cellular membranes, an important subject in biomembrane research with vast implications since the understanding the behavior of a large variety of proteins in mammalian can benefit a wide community in the life sciences.

This is an issue that has challenged researchers since it was first realized that different lipids and proteins are segregated into distinct sorting pathways, when studying polarized epithelial cells (late 80s). When the lipid raft model was proposed (1997), generalizing this concept to non-polarized epithelial cells, the interest in this subject was renewed and expanded. After this iconic paper by Simons and Ikonen the awareness of the importance of liquid ordered (Ld)/ liquid disordered (Lo) phase separation provided the biophysical framework within which the preference of proteins for lipid rafts versus non-raft domains should be addressed, i.e., the Lo/Ld (raft/non-raft) partition coefficient.

Despite several attempts, and the emergence of a few general rules over the years, a general quantitative model able to predict the raft preference of all transmembrane proteins has not been proposed yet. In this sense, the work here presented is quite pertinent and relevant.

In this work, the Authors address this problem making use of a recently established model that, in principle, should be close to a native plasma membrane, giant plasma membrane vesicles (GPMVs). They have designed a large number of constructs to systematically change several properties of the protein transmembrane segments, and have confirmed its length and degree of palmitoylation as determinant features that contribute to the overall raft preference. In Figure 3, Caption; Panel A, Line 2, the Authors acknowledge that "TMD length and palmitoylation have been previously identified as determinants of raft partitioning." This means that this part of the study could be regarded as mainly confirmatory. They also show that a much less exploited feature is also important for such preference, which is the surface area (solvent exposed surface), definitely an added value to their work.

Major concerns

Despite its virtues, there are some apparent limitations in this study, which the Authors should address:

1. The use of GPMVs – although retaining many features of the plasma membrane, in these vesicles, microscopic domains are observed, which were not detected in the plasma membrane, meaning that its organization is not fully retained. The Authors cite in the Introduction important references concerning confinement zones detected by single particle techniques. However, this confinement is highly dependent on the interactions of transmembrane proteins with the cytoskeleton and most of these interactions are dismantled in GPMVs.
2. The study is restricted to single-TMD proteins.
3. In this study, it is assumed that the raft preference is due to features of the TMD, and the possible role of extramembrane regions is not considered.
4. The important role of oligomerization is not addressed. Did the Authors analyse the tendency of the different sequences to form dimers, or higher order aggregates? (e.g., oligomerization enhances LAT preference for lipid rafts (ref. 18 in the MS)). A condensed phase such as Lo may facilitate aggregation which will affect the values retrieved for ΔG_s , K_p s, and exposed areas.
5. The study would gain considerable strength if, for at least a couple of cases (such as done for the simulations) controls in experimental model systems would be performed, in order to control and change quantitatively the protein: lipid ratio, and the fraction of Lo and Ld.
6. On page 4; Paragraph 1; Lines 1-2 the Authors state that “the large majority of PM proteins are not raft associated is consistent with previous estimates)”. However, it seems difficult to comply with the recent notion that the PM could be actually mainly in Lo state (e.g., ref. 8 in the MS; V. Kilin; Biophysical Journal Volume 108 May 2015 2521–2531; Fluorescence Lifetime Imaging of Membrane Lipid Order with a Ratiometric Fluorescent Probe.). This issue should also be addressed.
7. To demonstrate the validity of their model, the Authors present several experiments where they decrease the raft affinity of a given set of proteins, for instance, in Figure 4 Panels (E-G). The proof of concept should also include experiments where plasma membrane proteins which are not raft-associated, or do not exhibit any affinity for rafts, become more “raftophilic” through modifications of the structure related to the parameters studied.

Minor Concerns

Abstract

Page 1; Paragraph 1; Line 4 - In the sentence “Reasoning that protein transmembrane domains...” instead of reasoning it would be more appropriate to write “assuming”.

Introduction

Page 1; Paragraph 2; Lines 2-3 – The sentence “These domains result from attractive interactions (...) and repulsive interactions” could be misleading. Do the Authors mean that they result from enthalpic and entropic balance?

Page 1; Paragraph 2; Line 9 – Is it meant rafts or domains in general? in the sentence “strong evidence supporting their physiological...”).

Page 1; Paragraph 4; Line 7 – The meaning of near native membrane environment should be clarified; see major concerns.

Results and discussion

Page 2; Paragraph 1; Line 2- The word “key” is perhaps too strong.

Page 2; Paragraph 1; Lines 8-9-“partitioned preferentially” The wording “equally” or “slightly preferentially” could be more adequate.

Page 2; Paragraph 1; Line 10 - “showed much smaller raft affinity” instead of negligible. A 4.fold preference for non-raft domains could still mean that most protein is in the Lo phase if this comprises the majority of the plasma membrane.

Page 2; Paragraph 1; Line 17 – Possibly reword to “suggesting that” instead of “confirming that”.

Page 2; Paragraph 4; Lines 7-8 – Please explain how the value of (0.6 pN/ nm) was obtained from ref. 26.

Page 3; Paragraph 2; Line 3- Please rephrase the sentence (“some TMDs clearly favor”). For some segments there was at most a slight preference for lipid rafts.

Page 3; Paragraph 3; Lines 6-7- calculation details in legend of fig S3” Information is insufficient for the non specialized reader.

Page 3; Paragraph 4; Lines 2-3 – “This effect can be rationalized by invoking the hydrophobic mismatch, or “mattress”, model 32” could perhaps be better stated as “This effect can be rationalized by invoking the hydrophobic mismatch principle, as it is shown here through the application of a simplified version of the mattress model”.

Page 3; Paragraph 4; Line 6- When the Authors state that “longer TMDs partitioning to thicker domains” – they also convey the message that this alone is not sufficient to overcome the energetic cost of inserting the protein in a more compact phase. There are studies (with both experimental and theoretical approaches) using the mattress model showing that when the hydrophobic thickness of the protein has a better matching with the more ordered phase, the protein actually stays close to the ordered/disordered interface (Dumas et al., 1997; Biophys. J. 73: 1940-1953), instead of the bulk ordered phase.

Page 4; Paragraph 1; Line 8 – “ref missing [ref 45]” (of the proteome45); note that ref. 43 is not cited.

Conclusion

Page 5; Paragraph 1; Line 8 – Please rephrase “isolated plasma membrane”. This can be mistaken by ultra-centrifugation isolation, instead of GPMV.

Materials and Methods

Page 5; Paragraph 3; Line 6 – Why where the GPMVs observed at 10°C? Cells were grown at 37°C (Page 5; Paragraph 2; Line 3). Can the Authors ascertain that at 10°C there are no gel domains?

Page 6; Paragraph 2; Line 3 – The Authors mention that, “For these simulations, diarachidonoyl PC (DAPC) was used instead of DLiPC because it yielded a more stable domain boundary”. Why did the Authors use DLiPC in some of the simulation studies, and not always DAPC?

References

Page 8; Reference 34 – Something is missing in the title: “L and L phases”.

Figures

Figure 1

Panel C - K_p is within error minor or equal to 1, therefore the Authors cannot state that a certain protein is raft preferring.

Caption; Panel B description; Lines 5-6; - “eliminates the native raft affinity” should be replaced by “decreases the relative raft-affinity”.

Caption; Panel C description; Lines 8 – The sentence, “suggesting a distributed feature” sounds incomplete; maybe add “responsible for the higher K_p ”.

Figure 2

General comment- Is the expression level each of studied protein identical in every experiment? How was that controlled? In case it was not, what implications could that have for the conclusions (e.g., more or less aggregation for different proteins – see major concerns)?

Figure 3

General comment- It is suggested that the Authors try to increase the raft affinity of LDLR, by either increasing its TMD length or palmitoylation.

Panel C- Can the Authors estimate % predictability for a given TMD? Since this model does not account for proteins with multiple TMD, the model can be highly limited in this regard.

Caption; Panel D description; Line 9- LIME instead of LIM

Caption; Panel F description; Line 13 – In the sentence “truncating the TMD length by 6 amino acids reduced raft partitioning for LAT, PAG, and LIME”, it should be indicated that this did not occur in all cases shown.

Figure 4

Caption; Panel C description; Line 4- After “PM proteins” it should be added “with a single TM segment”.

Caption; Panel D description; Line 6 – Instead of “clearly not normal”, perhaps replace by “clearly not unimodal”.

Reviewer #4 (Remarks to the Author):

In this paper the author propose a simple model for how single spanning membrane proteins attach to lipid drafts. The authors basically propose that the affinity is related to the surface area of the sidechains and not to particular sequence features. The authors show this convincingly for a small number of experimental peptides.

Although this problem has been studied somewhat before I do find the paper interesting. However, the value of the paper would significantly be increased if a more through analysis of the sequence features that effect lipid draft was analyzed. In particular I do think you could obtain equal good correlations by using average hydrophobicity as with ASA, i.e. it might not really be the surface area but hydrophobicity that is important. This should be tested.

Further it would be interesting to analyze how the length of the hydrophobic region effects the results.

We sincerely thank the editor and reviewers for their constructive critiques of our manuscript “Structural determinants and functional consequences of protein affinity for membrane rafts” (manuscript #: NCOMMS-16-26334-T). We have revised the manuscript in accordance with reviewer concerns, including four new experimental figures and extensive revision of the manuscript. Most notably, we have emphasized the novelty and cellular relevance of our findings, which we believe to be a strength of this manuscript. Additionally, we have experimentally addressed reviewer concerns regarding the role of TMD roughness and hydrophobicity, with the results strengthening our original conclusions.

We believe that the reviewers’ comments, and the experimental evidence generated in response to them, have dramatically improved the manuscript, and that it is now acceptable for publication in *Nature Communications*.

Below please find a point-by-point response to the reviewer comments.

Reviewer #1 (Remarks to the Author):

The paper by Lorent et al. describes a simple physical model applicable to transmembrane proteins based on 3 measurable parameters: number of palmitoylation cysteines, length of transmembrane domain and surface area of the side chains. Such model allows the calculation of the Gibbs free energy change upon partitioning from lipid disordered to ordered phases in the plasma membrane. More precisely, the model is derived by measuring partitioning of specifically designed polypeptides between liquid-disordered and liquid-ordered phases in vesicles artificially induced from the plasma membrane (GPMV). The model is also used to predict the partition behavior in GPMV for 729 single-pass transmembrane proteins from the human proteome. Finally, the authors show that, from a practical point a view, their model can be used to predict protein distribution among different cellular membranes (i.e. plasma membrane, ER or Golgi). The manuscript is very well written and the conclusions are compatible with the experimental results.

We thank the reviewer for their careful reading of the manuscript and complimentary comments.

1-1. I have few comments regarding the experimental part and the clarity of exposition. When K_p is measured for the different constructs, is the expression of such constructs comparable? Can K_p be a function of protein concentration?

The TMD constructs are all designed from the same backbone and promoter, thus should have comparable expression levels. This was validated in a previous publication¹. Furthermore, we examined whether there was a dependence of partitioning on PM concentration (as quantified by RFP intensity) and found no notable dependence for any of the constructs tested. These data are now included in Fig S5 and reproduced below.

Fig S5. No notable correlation was observed between expression level and raft partitioning for either raft- or non-raft-preferring TMDs.

1-2. The captions should better explain the difference (e.g. overlapping) between the data in Figs. 2A and 3B

This has been added to the caption of Fig 3B.

1-3. In the third paragraph of the results, the authors state that “some TMDs clearly favor the raft phase”. Nevertheless, in the results, I do not see any reported ΔG of partition clearly lower than zero (or K_p significantly larger than 1). The only exception is in Figure 1D. Regarding this figure though, why is the WT K_p larger than 1 in panel D but lower (for the same WT) in panel C?

This was inaccurate wording on our part. The reviewer is correct that the highest raft affinities ($K_{p,raft}$) we observe are near unity, i.e. these constructs do not strongly favor the raft phase, but rather have approximately equal partitioning between the two phases. The intended meaning was that some TMDs partition to the raft phase while others do not; the manuscript has been changed to reflect this.

The minor differences in the partitioning of WT LAT between panels C and D reflect the typical variation in absolute $K_{p,raft}$ values in these experiments. We measure WT LAT partitioning in every experiment as a control to ensure consistency.

1-4. In the third paragraph of the results, the authors state that the model “yields excellent agreement” with the observed partitioning. Nevertheless, Figure 3B shows a cluster of circa 20 proteins (out of circa 60) showing an overestimation of the predicted ΔG .

The reviewer is correct in pointing out the cluster of TMDs whose raft phase affinity our model underestimates by ~ 0.2 kcal/mol. We noticed this set of TMDs and have been actively exploring the origin of this effect. Our working hypothesis is that residues present in the WT LAT TMD and some of these slight outliers produce weak effects that slightly affect partitioning. These effects could include subtle helix kinks/tilts, weak intermolecular interactions, or slight affinity for certain membrane lipids. This possibility is consistent with the excellent fit we observe to the exclusively Leu/Ala containing TMDs in Fig 2C, in which such effects would not be present. These questions will be explored in future work. Most importantly, such effects, if present, are relatively weak contributors to raft affinity, as shown by the very good agreement between the ASA model and the data in Fig 2A-B for over 60 constructs. For this manuscript, we have changed the wording of that sentence and pointed out the cluster.

1-5. Regarding the role of the exposed surface area: Could there be a role played by the roughness of the surface rather than just the amount of the exposed surface? A smooth/flat surface might have more favorable interactions with the ordered lipid environment. Maybe a role of a surface-volume ratio?

This is a very good suggestion, which we have carefully explored. It is a conceptually tricky problem because there is a coupling between side-chain surface area and surface roughness in transmembrane helices, depending on the spatial dimensions being probed. For example, very small probes can freely explore the entire amino acid side-chain, and thus side-chain surface areas will be perfectly correlated with “roughness” values at this length scale. In contrast, very large probes will not detect even large crevices between adjacent side chains. Thus, the choice of length scale, and probing multiple length scales, to determine roughness is crucial.

To address this issue, we computationally calculated TMD roughness via fractal dimension² using the MSMS algorithm³, for spherical probes between 5 and 6 Å. This length scale was chosen because it has been estimated that the persistence length of a phospholipid acyl chain in a fluid bilayer is ~ 5 Å⁴, and this persistence length would determine the length scale of features that could be “sensed” by lipids in a bilayer.

Applying this calculation to the Ala/Leu TMDs used for Fig 2C, we observe a weak correlation between TMD roughness and raft phase partitioning, with $R^2 = 0.37$, compared to $R^2 = 0.92$ for the correlation between surface area and partitioning (Fig 2C). The weak correlation between roughness and raft affinity is likely due to the fact that surface roughness and side-chain surface area are inherently coupled for TMDs, even at this length scale.

This discussion has been included in the manuscript and as Fig S3, also reproduced below.

Fig S3. Weak correlation between raft affinity and TMD surface roughness calculated via the fractal dimension for probes between 5-6 Å. At this scale, all TMDs tested appear nearly smooth (fractal dimension = 2 for an ideal smooth surface).

1-6. Similarly, the data in Figure 2 A show that (differently from what predicted by the model) TMDs with a surface area below 750 Å² have a very low deltaG of partition, independently from the actual surface area value. Can the prediction be improved in this case by including the TMD length term?

This population in Fig 2A is the same one from Fig 3B. The nominal length of the TMD is not a factor in these results, as all constructs have the same number of amino acids. However, as discussed in the answer to point 4 above, other features of the TMDs may be slightly affected by specific residues of individual constructs. These possibilities will be explored in future work.

1-7. At the end of the third paragraph, the author state that they could rationally design a “raft preferring” LDLR protein. Nevertheless, this appear once again confusing since the relative K_p is not significantly higher than 1. Also at this regard – just a small note -, choosing a protein candidate from the lower part of figure 3C might result in more impressive results for figure 3D

This is a good point, also addressed in the answer to question 1-3 above. The wording has been revised as above.

The cluster of proteins tested to represent ‘predicted raft proteins’ was intentionally chosen to have similar predicted raft partitioning as our ‘seed’ for building the model (i.e. LAT), to avoid biasing our validations by testing only the most ‘raft preferring’ candidates. We are currently investigating partitioning of other proteins more broadly.

1-8. On a different note, I am concerned about the originality of the conclusions and how this study can appeal a vast amount of general readers. First, even for scientists working in the field of cell biology, the concept (i.e. the importance in a biological context) of raft domains is not accepted in general. For each study proving existence of raft domains mentioned by the authors , there is a study arguing against the general definition and existence of raft domains (as they are usually defined, e.g. Frisz et al. JBiolChem 2013) or their role in cell biology (e.g. Wilson et al. BiophysJ 2015). Without going too much into details here, also the authors state that this is a controversial topic.

We agree that the biological functions of rafts remain an area of active research and debate, and the original version of the manuscript did not accurately represent the controversy. This has now been rectified. However, we would disagree with the characterization that there is equal weight of evidence for and against the existence of membrane domains. The self-organizing properties of mammalian membrane lipids are undeniable, and the formation of liquid-ordered and liquid-disordered phases in GPMVs confirms the capacity for such organization in a complex, protein-rich membrane derived from live cells. Such phase separation in GPMVs has recently been related to membrane structure in live cells, with elegant experiments from independent labs showing that fluorescent lipid analogs that partition to Lo domains in GPMVs also show distinct diffusive behaviors in live cells^{5,6}, strongly suggesting the presence of nanoscopic Lo domains in live cells.

We are aware of the exciting work with nanoSIMS and the conclusions regarding SM- (but not cholesterol-) enriched domains and have now included citations to it. In our view, these results do not necessarily argue against the presence of ordered domains in biological membranes, since highly enriched sphingomyelin domains should be more tightly packed than the surrounding bulk membrane, even in the absence of dramatic cholesterol enrichment. Indeed, Feigenson and Buboltz have suggested that L_o and L_d phases in the DPPC/POPC/Chol system may have very similar cholesterol levels⁷. Importantly, our model is independent of the actual compositions of the coexisting domains, rather only that there is a difference in lipid packing, which has been definitively shown in GPMVs⁸⁻¹¹.

Ultimately, the putative features of membrane domains suggested by a variety of techniques – i.e. nanometric, dynamic, relatively similar in composition to the surrounding membrane, potentially diverse/heterogeneous – make them inherently difficult to study in their native context. This is one reason why we believe studies like ours, which utilize a natural model system to directly quantify raft-associated behaviors and relate them to cellular effects (in our case, protein sorting), are essential for moving this field forward.

1-9. The model presented is validated by (and allows to predict mostly) the outcome of experiments in which partition of proteins is determined between artificial lipid phases in membrane system whose biological relevance is at least questionable. How useful is it to predict whether a certain protein will partition in the ordered phase of a GMPV or not?

GPMVs are cell-derived membranes with natural diversity and complexity of lipids and proteins. Spontaneous formation of coexisting fluid domains in these GPMVs demonstrates the self-organizing capacity of mammalian plasma membranes, and the potential for such organization to laterally sort membrane proteins. GPMVs are not identical to live cell PMs, and the potential caveats associated with them have been described by us in previous work and have now also

been highlighted in the manuscript. Despite these caveats, a number of studies support the cellular relevance of phase separation in GPMVs and the partitioning of proteins therein^{5,6,12,13}. Most notably, the above-referenced work showing the close relationship between Lo partitioning and diffusion behavior^{5,6}, but also recent observations that proteins which partition to the ordered-phase in GPMVs form raft-like nanoscopic clusters in intact cells and that these behaviors are important for immune cell function¹².

1-10. How useful is it to predict whether a certain protein will partition in the ordered phase of a GMPV or not? If the authors had showed instead that the model allows the prediction of a generally relevant and measurable outcome (e.g. involvement of a protein in a signaling pathway), the study would have had a much higher relevance.

We believe the above-cited studies provide evidence that raft phase partitioning in GPMVs is related to diffusion and function in live cells. Further, a major reason for the historical controversy surrounding the raft field is that non-quantitative and unreliable methodologies have prevented consistent definition of which proteins are supposed to be present in lipid rafts and why. Our work directly addresses this limitation by direct, quantitative measurements in a biological membrane, supported by coherent physical models. The ultimate goal of defining the determinants of lipid raft partitioning is to generate robust predictions for raft affinity that may help resolve some of the major controversies in the field. Most importantly, our model relates raft partitioning to a relevant and measurable cellular outcome, specifically subcellular protein localization, as discussed below.

1-11. Second, the author do indeed provide evidence that their model allows also the prediction of more general and interesting features, namely intra-cellular membrane distribution of proteins and their partition in the plasma membrane. This section of the results is of course much more relevant. I find though that this part is not entirely original and innovative: Intracellular sorting of membrane proteins predicted by TMD length and palmitoylation has been reported in the past (e.g. Chum et al. J of Cell Science 2015). Sharpe et al. (Cell, 2010) proposed a model that uses the primary structure of a transmembrane protein to predict whether it will be found in the (asymmetric raft-like) bilayer environment of the plasma membrane.

We thank the reviewer for pointing out this important point, which was not adequately emphasized in the original submission. The reviewer is correct in pointing out that features of TMDs have been previously related to sub-cellular localization. However, those studies did not define a mechanistic explanation for their observations. Indeed, the paper of Sharpe et al¹⁴ proposes two possible mechanisms to explain their biostatistical observations. One mechanism suggests that protein TMDs pull lipids along to create the distinct lipid composition of the PM and other cellular organelles, while the other proposes raft-like lipid domains that recruit specific proteins and deliver them to their cellular location. Ultimate Sharpe et al conclude that “determining the relative contributions of proteins and lipids to each other’s sorting is likely to be a key issue for future studies of the biogenesis of eukaryotic membranes.”

Our study resolves this question by direct experimental evidence in support of the latter hypothesis, by showing that the protein features that determine raft affinity (TMD length, surface area, and palmitoylation) are the very same ones identified by Sharpe et al as determinants of PM localization. Therefore, our observations strongly suggest that for certain proteins, PM localization is driven by their raft association, and therefore that raft-mediated sub-cellular sorting/trafficking is the reason that long, thin TMD-containing proteins are enriched at the PM. We experimentally support this inference by showing that mutation of any of the raft-targeting features in two different proteins leads to reduced PM localization. These data confirm the statistical observations of Sharpe et al by direct experimental evidence. Most importantly, they provide strong experimental evidence in support of a mechanistic explanation for the important and exciting previous observations.

We have now included an extended discussion of the findings of Sharpe et al and the mechanistic explanation provided by our observations.

1-12. In conclusion, this study is undoubtedly of very high quality and it clarifies in a quantitative way some ideas that were circulating since some time (in the circle of people working with the model of raft/liquid ordered domains in the plasma membrane). I only doubt whether this study might be of general interest in cell biology.

Regarding general interest of this concept to cell biologists, the frequency of published manuscripts archived by Pubmed for the search term “lipid raft” has not decreased in ~15 years (~200/year), despite the proliferation of alternative terms such as microdomains/nanoclusters for similar or related concepts. A 2010 review on the subject¹⁵ has been cited >1800 times. More specifically, our previous work on raft partitioning in GPMVs¹⁶ has been cited 186 times since 2010. We believe these facts support the continuing interest and relevance of this concept. This broad interest justifies, and even demands, continued investigations to refine the concept and eliminate the persistent uncertainty around a topic of clear importance to the community.

Specific to our study, the delineation of ‘rules’ that govern protein raft affinity will help refine the cellular roles of lipid rafts by clarifying the raft proteome. Finally, our conclusions suggest that trafficking via ordered membrane domains is a mechanistic explanation for a fundamental cellular design principle, one that is conserved from yeast to mammals, that different organelles possess distinct protein TMD features. We believe these conclusions will be of broad interest and impact to both the cell biology and biophysics communities.

Reviewer #2 (Remarks to the Author):

This paper links features of transmembrane helices to their propensity to partition into ordered parts of giant plasma membrane vesicles, and extends these observations to larger classes of proteins and a physical model that describes partitioning at the level of differential surface tensions. This is quite intriguing and supported by a significant amount of data. I have no major comments and it will be interesting to see the scientific community follow up on the ideas in this paper.

We appreciate the complementary comments.

Reviewer #3 (Remarks to the Author):

The work by Leventhal and col. tackles in a systematic manner and with unprecedented detail the structural features that account for single transmembrane domain (TMD) proteins preference for lipid raft domains in mammalian cellular membranes, an important subject in biomembrane research with vast implications since the understanding the behavior of a large variety of proteins in mammalian can benefit a wide community in the life sciences. This is an issue that has challenged researchers since it was first realized that different lipids and proteins are segregated into distinct sorting pathways, when studying polarized epithelial cells (late 80s). When the lipid raft model was proposed (1997), generalizing this concept to non-polarized epithelial cells, the interest in this subject was renewed and expanded. After this iconic paper by Simons and Ikonen the awareness of the importance of liquid ordered (Ld)/ liquid disordered (Lo) phase separation provided the biophysical framework within which the preference of proteins for lipid rafts versus non-raft domains should be addressed, i.e., the Lo/Ld (raft/non-raft) partition coefficient. Despite several attempts, and the emergence of a few general rules over the years, a general quantitative model able to predict the raft preference of all transmembrane proteins has not been proposed yet. In this sense, the work here presented is quite pertinent and relevant.

We thank the reviewer for their thoughtful comments and appreciation of the relevance of our work.

In this work, the Authors address this problem making use of a recently established model that, in principle, should be close to a native plasma membrane, giant plasma membrane vesicles (GPMVs). They have designed a large number of constructs to systematically change several properties of the protein transmembrane segments, and have confirmed its length and degree of palmitoylation as determinant features that contribute to the overall raft preference. In Figure 3, Caption; Panel A, Line 2, the Authors acknowledge that “TMD length and palmitoylation have been previously identified as determinants of raft partitioning.” This means that this part of the study could be regarded as mainly confirmatory. They also show that a much less exploited feature is also important for such preference, which is the surface area (solvent exposed surface), definitely an added value to their work.

TMD length and palmitoylation were the focus of our previous publications. In this manuscript, these previously identified principles were used towards building a comprehensive, predictive model for raft partitioning of single-pass proteins.

Major concerns. Despite its virtues, there are some apparent limitations in this study, which the Authors should address: 3-1. The use of GPMVs – although retaining many features of the plasma membrane, in these vesicles, microscopic domains are observed, which were not detected in the plasma membrane, meaning that its organization is not fully retained. The Authors cite in the Introduction important references concerning confinement zones detected by single particle techniques. However, this confinement is highly dependent on the interactions of transmembrane proteins with the cytoskeleton and most of these interactions are dismantled in GPMVs.

GPMVs are indeed a model system for the live cell plasma membrane and do not faithfully represent certain features thereof. These caveats have been detailed by us in previous publications, and we have now added a discussion to the present manuscript (penultimate paragraph of Introduction). Nevertheless, a number of recent studies support the cellular relevance of phase separation in GPMVs and the partitioning of proteins therein^{5,6,12,13}. Most notably, elegant experiments from independent labs showed that fluorescent lipid analogs that enrich in Lo domains in GPMVs are also

detergent-resistant (the biochemical hallmark of raft affinity), and most importantly show distinct diffusive behaviors in live cells^{5,6}. These findings are consistent with the presence of nanoscopic Lo domains in live cells, which are manifested as large Lo phases in GPMVs. Another set of observations showed that proteins which partition to the ordered-phase in GPMVs form raft-like clusters in intact cells and that these behaviors are important for immune cell function¹². This discussion has also been added to the Introduction.

One of the major differences between live cell PMs and GPMVs is indeed the presence of a densely associated cytoskeletal network (this is now noted in the manuscript). Interestingly, there is accumulating evidence¹⁷⁻²⁰ that suggests this membrane-associated cytoskeleton is the reason that live cell PMs do not separate into microscopic domains, as GPMVs do. The hypothesis is that proteins and lipids immobilized by their association with the cytoskeleton prevent large-scale phase separation, instead restricting lipid-driven domains to nanoscopic scale in live cells. This domain fragmentation is certainly not the only effect of actin-bound proteins on PM biophysics (e.g. actin corrals), but it likely contributes to actin-dependent effects on single-molecular diffusion. We have carefully reviewed our references and replaced some inappropriate ones (specifically about SPT tracking) with those more directly implicating Lo domains in single-molecule diffusion.

Importantly, the over-arching conclusion from a variety of observations (including but not limited to those cited above, as well the data presented in this manuscript) is that Lo-like domains are present as nanoscopic entities in membranes of live cells and that these influence aspects of cellular physiology, including signaling and trafficking. We believe these conclusions, and the difficulty inherent to studying functional membrane organization in live cells, justify the relevance of studying protein partitioning to macroscopic Lo domains in GPMVs.

3-2. The study is restricted to single-TMD proteins.

Single-pass transmembrane proteins (TMPs) represent ~50% of all TMPs²¹ and ~15% of the mammalian proteome²², justifying the relevance of studying them. Prior to our work, there have been no generalizable insights into raft partitioning of TMPs, thus we feel it is appropriate to begin with the simplest, though large, subset of TMPs in order to make progress. Multipass TMPs present the unique challenge that not all TMD residues will be in contact with surrounding lipids, making *de novo* prediction of even simple parameters like ASA extremely challenging. We are currently designing strategies to address this challenge. In the manuscript, we have clarified wherever necessary that our study is restricted to single-pass TMPs.

3-3. In this study, it is assumed that the rat preference is due to features of the TMD, and the possible role of extramembrane regions is not considered.

This is correct, the aim of our study is to test how features of the transmembrane domain influence plasma membrane localization and raft partitioning. However, we have previously shown that raft partitioning for a subset of proteins is solely dependent on their TMDs¹. For one of these proteins (i.e. LAT), we tested a minimal construct that contained no amino acids outside of the TMD; this construct partitioned identically with the full-length protein¹. Finally, the data in Fig 3E-G confirm the predicted effects of TMD mutations on raft partitioning, suggesting that at least for these proteins, the TMD is the determinant of partitioning.

However, we agree that extramembrane regions are likely to be important for partitioning in certain proteins and have added a note to the manuscript regarding this possibility.

3-4. The important role of oligomerization is not addressed. Did the Authors analyse the tendency of the different sequences to form dimers, or higher order aggregates? (e.g., oligomerization enhances LAT preference for lipid rafts (ref. 18 in the MS)). A condensed phase such as Lo may facilitate aggregation which will affect the values retrieved for ΔG_s , K_p s, and exposed areas.

This is a good point, and we have also strongly considered a potential role of oligomerization. We agree that dimerization could increase L_o phase partitioning and that Lo partitioning might increase dimerization. As the reviewer correctly points out, we have previously shown a robust effect of enhanced raft affinity upon exogenously induced dimerization of LAT. Because those results show that dimerization *enhances* raft affinity, and here we did not observe any constructs with significantly greater raft affinity than WT LAT, we conclude that the TMDs produced here did not *induce* significant dimerization.

Also, we point out that some of the constructs tested here have no native LAT TMD residues (and thus that any putative dimerization sequence would be mutated) yet still show robust raft-partitioning, at parity with WT LAT.

Further, we attempted to determine whether WT LAT is dimerized/oligomerized by FLIM-FRET, chemical crosslinking, native PAGE, and FCS (via FCCS and number/brightness analysis), and whether the mutants we created reduced that dimerization. All of these methods definitively rule out the presence of stable LAT dimers/oligomers, as we observed no strong evidence of dimerization. This is consistent with the absence of literature suggesting autonomous LAT oligomerization. Furthermore, we observed no significant differences using any of these methods between trLAT-WT and trLAT-allL, which have vastly different partitioning values. Unfortunately, none of these experimental methods can definitively rule out weak interactions that could yield transient/inabundant dimerization, thus we feel it is inappropriate to conclusively report that LAT is strictly monomeric.

Most importantly, newly generated data strongly argues against any effect of oligomerization on the results reported in the present manuscript. Specifically, we observe no correlation between the expression level of three different constructs (with different partitioning values) and their raft partitioning (Fig S7). Based on mass action, any potential oligomerization would be strongly affected by concentration. The fact that we observe no such dependence despite >10-fold differences in protein concentration is strong evidence that oligomerization has no effect on our observations, either because there is no oligomerization, or (less likely) because oligomerization does not affect partitioning.

3-5. The study would gain considerable strength if, for at least a couple of cases (such as done for the simulations) controls in experimental model systems would be performed, in order to control and change quantitatively the protein: lipid ratio, and the fraction of Lo and Ld.

We agree with the reviewer and this is why we believe the computational data in Fig 2D-F is an important validation of the principle we report. Unfortunately, controls in experimental model systems are problematic because partitioning in synthetic model systems does not recapitulate the behavior in isolated GPMVs. This disparity has been reported for a variety of lipid dyes¹⁰, but more importantly, it has been directly observed for LAT. A peptide based on the LAT TMD, with palmitoylated cysteine residues, is completely excluded from Lo phases in synthetic GUVs²³, whereas it (slightly) enriches in the Lo phase in GPMVs. As we have previously discussed²⁴, we believe this disparity is due to the exaggerated interdomain differential in lipid packing in GUVs, which have “unnaturally” high packing in the Lo phase and low packing of the Ld phase, compared to GPMVs⁸. Practically, the artefactual exclusion of LAT (and indeed all other transmembrane proteins) from the Lo phase in GUVs makes this model prohibitive for examining the determinants of partitioning between coexisting phases, making the GPMV approach a key addition to the field.

However, we believe that our previous experiments address some of the reviewer’s concern about the robustness of our findings. First, as demonstrated in Fig S7, the abundance of our target protein and its relative ratio to lipids does not affect partitioning. Second, we have previously demonstrated that the partitioning of LAT and several variants is identical in GPMVs made from two unrelated cell lines, which almost certainly have different lipid compositions and membrane properties¹. Third, we have shown that treatments that dramatically affect GPMV physical properties have minimal effects on partitioning⁹. Specifically, deoxycholic acid reduces the packing of the Ld phase and increases the phase separation temperatures in GPMVs by up to 20 °C. Despite these changes, the partitioning of LAT and several other proteins is not significantly affected. Therefore, we believe our observations are robust against physiologically relevant changes in membrane properties.

3-6. On page 4; Paragraph 1; Lines 1-2 the Authors state that “the large majority of PM proteins are not raft associated is consistent with previous estimates)”. However, it seems difficult to comply with the recent notion that the PM could be actually mainly in Lo state (e.g., ref. 8 in the MS; V. Kilin; Biophysical Journal Volume 108 May 2015 2521–2531; Fluorescence Lifetime Imaging of Membrane Lipid Order with a Ratiometric Fluorescent Probe.). This issue should also be addressed.

We agree with the notion that a large fraction of the PM (perhaps a majority) is in the Lo state. However, we do not believe this is incompatible with our proposal that the majority of PM proteins are not raft associated. Previous observations in GPMVs²⁵ and DRMs²⁶ suggest that ordered/raft environments are relatively protein poor compared to disordered/non-raft. This is consistent with our inference that the relatively higher surface tension between the Lo phase and protein TMDs resulting from greater lipid-lipid interactions would tend to exclude most proteins.

3-7. To demonstrate the validity of their model, the Authors present several experiments where they decrease the raft affinity of a given set of proteins, for instance, in Figure 4 Panels (E-G). The proof of concept should also include experiments where plasma membrane proteins which are not raft-associated, or do not exhibit any affinity for rafts, become more “raftophilic” through modifications of the structure related to the parameters studied.

This is an excellent suggestion, and we have included data in Fig 3G that shows exactly this effect. According to our predictions, the TMD of LDLR should have very low affinity (Fig 3C), and this is indeed what we observed (Fig3D).

By reducing the ASA of the TMD and adding palmitoylation sites, we were able to increase raft affinity of the LDLR TMD to almost the same level as our highest raft-partitioning proteins (e.g. LAT) (Fig 3G). We believe this is a key proof-of-concept confirmation.

Minor Concerns

Abstract

Page 1; Paragraph 1; Line 4 - In the sentence “Reasoning that protein transmembrane domains...” instead of reasoning it would be more appropriate to write “assuming”.

We have changed to “Hypothesizing”.

Introduction

Page 1; Paragraph 2; Lines 2-3 – The sentence “These domains result from attractive interactions (...) and repulsive interactions” could be misleading. Do the Authors mean that they result from enthalpic and entropic balance? Not exactly. We mean attractive interactions like those between saturated lipids and cholesterol, and repulsive interactions like those between saturated and polyunsaturated lipids. However, we agree with the reviewer, this is clumsy wording, since all lipids attract each other in aqueous conditions. We have changed this wording to “relatively preferred interactions”.

Page 1; Paragraph 2; Line 9 – Is it meant rafts or domains in general? in the sentence “strong evidence supporting their physiological...”.

We intended this to be limited to rafts specifically, defined as lipid-driven relatively ordered domains. We have removed references which were inappropriate for such a meaning, namely to single-particle tracking experiments that rather implicated actin corrals.

Page 1; Paragraph 4; Line 7 – The meaning of near native membrane environment should be clarified; see major concerns.

Agreed, and we have added our discussion of GPMV caveats here.

Results and discussion

Page 2; Paragraph 1; Line 2- The word “key” is perhaps too strong.

Agreed, changed to “important”.

Page 2; Paragraph 1; Lines 8-9- “partitioned preferentially” The wording “equally” or “slightly preferentially” could be more adequate.

Agreed, changed as suggested.

Page 2; Paragraph 1; Line 10 - “showed much smaller raft affinity” instead of negligible. A 4 fold preference for non-raft domains could still mean that most protein is in the Lo phase if this comprises the majority of the plasma membrane.

Agreed, changed as suggested.

Page 2; Paragraph 1; Line 17 – Possibly reword to “suggesting that” instead of “confirming that”.

Agreed, changed as suggested.

Page 2; Paragraph 4; Lines 7-8 – Please explain how the value of (0.6 pN/ nm) was obtained from ref. 26.

That reference (Risselada et al; Fig 4D) estimates the surface tension between Lo and Ld domains at 0.15 kT/nm², which is 0.62 pN/nm. This explanation has been added to the manuscript.

Page 3; Paragraph 2; Line 3- Please rephrase the sentence (“some TMDs clearly favor”). For some segments there was at most a slight preference for lipid rafts.

Agreed, changed to “partition to the raft phase”.

Page 3; Paragraph 3; Lines 6-7- calculation details in legend of fig S3” Information is insufficient for the non specialized reader.

We have extended and clarified this explanation in the Supplement.

Page 3; Paragraph 4; Lines 2-3 – “This effect can be rationalized by invoking the hydrophobic mismatch, or “mattress”, model 32” could perhaps be better stated as “This effect can be rationalized by invoking the hydrophobic mismatch principle, as it is shown here through the application of a simplified version of the mattress model”.

Agreed, changed as suggested.

Page 3; Paragraph 4; Line 6- When the Authors state that “longer TMDs partitioning to thicker domains” – they also convey the message that this alone is not sufficient to overcome the energetic cost of inserting the protein in a more compact phase. There are studies (with both experimental and theoretical approaches) using the mattress model showing that when the hydrophobic thickness of the protein has a better matching with the more ordered phase, the protein actually stays close to the ordered/disordered interface (Dumas et al., 1997; Biophys. J. 73: 1940-1953), instead of the bulk ordered phase.

This is an interesting point, but the referenced manuscript rather concerns the behavior of a multipass membrane protein (a GPCR) in a membrane that shows liquid-gel phase coexistence, whereas our work solely focuses on single-pass membrane proteins in L_o/L_d membranes. It is likely that the tight packing of the gel phase is almost entirely prohibitive to protein insertion, especially a large helix bundle like a GPCR, and thus the free energy of transferring a protein from the L_o to the gel phase may be very high. In contrast, the coexisting liquid phases in GPMVs are relatively similar with respect to lipid packing⁸, making it relatively less costly to introduce small inclusions like a single α -helix. Ultimately, in all our experiments, we observe homogeneous fluorescence signal over the entire phase and no enrichment at the phase boundaries.

Page 4; Paragraph 1; Line 8 – “ref missing [ref 45]” (of the proteome45); note that ref. 43 is not cited.

We thank the reviewer for their keen eye; there was a mis-numbering of the references that has now been corrected.

Conclusion

Page 5; Paragraph 1; Line 8 – Please rephrase “isolated plasma membrane”. This can be mistaken by ultra-centrifugation isolation, instead of GPMV.

Agreed, changed to “isolated plasma membrane vesicles”.

Materials and Methods

Page 5; Paragraph 3; Line 6 – Why where the GPMVs observed at 10°C? Cells were grown at 37°C (Page 5; Paragraph 2; Line 3). Can the Authors ascertain that at 10°C there are no gel domains?

L_o/L_d phase separation in GPMVs is temperature-dependent, and large microscopic domains are only observed below the miscibility transition temperature (T_{misc}). 10C was chosen relatively arbitrarily as a temperature well below T_{misc} ; however, this is the temperature previously used by us and other for characterization of phase separated GPMVs. Incidentally, we have previously observed gel phase domains in GPMVs, but only upon wholesale cholesterol depletion

Page 6; Paragraph 2; Line 3 – The Authors mention that, “For these simulations, diarachidonoyl PC (DAPC) was used instead of DLiPC because it yielded a more stable domain boundary”. Why did the Authors use DLiPC in some of the simulation studies, and not always DAPC?

This choice was made for two reasons. First, we would have preferred DLiPC for both sets of studies since it is more similar to native PM lipids than DAPC and has been used more extensively in simulations. However, for the PMF calculations, the DLiPC-containing systems did not yield a suitably stable boundary, causing a lack of convergence in the umbrella sampling simulations. Second, changing the lipid system and observing the same trends gave us confidence in the robustness of the results.

References

Page 8; Reference 34 – Something is missing in the title: “L and L phases”.

We again thank the reviewer for their keen eye and have fixed the problem.

Figures

Figure 1 ; Panel C - K_p is within error minor or equal to 1, therefore the Authors cannot state that a certain protein is raft preferring.

Agreed, changed as suggested

Caption; Panel B description; Lines 5-6; - “eliminates the native raft affinity” should be replaced by “decreases the relative raft-affinity”.

Agreed, changed as suggested.

Caption; Panel C description; Lines 8 – The sentence, “suggesting a distributed feature” sounds incomplete; maybe add “responsible for the higher K_p ”.

Agreed, changed as suggested.

Figure 2 General comment- Is the expression level each of studied protein identical in every experiment? How was that controlled? In case it was not, what implications could that have for the conclusions (e.g., more or less aggregation for different proteins – see major concerns)?

The TMD constructs are all designed from the same backbone and promoter, thus should have comparable expression levels. This was validated in a previous publication¹. Furthermore, we examined whether there was a dependence of partitioning on PM concentration (as quantified by RFP intensity) and found no notable dependence for any of the constructs tested. These data are now included in Fig S7.

Figure 3 General comment- It is suggested that the Authors try to increase the raft affinity of LDLR, by either increasing its TMD length or palmitoylation.

The TMD of LDLR had low raft affinity despite the same number of transmembrane amino acids as LAT (one of the highest raft partitioning constructs) and the presence of two palmitoylation sites (see Fig 3D and Supp Table IV). Thus, we hypothesized that a small TMD ASA was necessary for its efficient raft partitioning. We validated this prediction by mutating several Leu and Phe residues to Ala, which reduced the TMD ASA and significantly increased raft partitioning, almost to the level of LAT (Fig 3G).

Panel C- Can the Authors estimate % predictability for a given TMD? Since this model does not account for proteins with multiple TMD, the model can be highly limited in this regard.

Although we accurately predicted all five of the TMDs tested in the article, it is unlikely that our model will be perfectly predictive for all possible TMDs, and it is certainly limited to single-pass TMPs. We have initiated experiments to test our predictions proteome-wide. The model as currently constructed is only valid for single-pass transmembrane proteins, and this has now been clarified in the manuscript. However, we again point out that single-pass TMPs comprise about half of all membrane proteins, making our predictions broadly useful.

Caption; Panel D description; Line 9- LIME instead of LIM

Caption; Panel F description; Line 13 – In the sentence “truncating the TMD length by 6 amino acids reduced raft partitioning for LAT, PAG, and LIME”, it should be indicated that this did not occur in all cases shown.

Agreed, changed as suggested.

Figure 4 Caption; Panel C description; Line 4- After “PM proteins” it should be added “with a single TM segment”.

Agreed, changed as suggested.

Caption; Panel D description; Line 6 – Instead of “clearly not normal”, perhaps replace by “clearly not unimodal”.

Agreed, changed as suggested.

Reviewer #4 (Remarks to the Author):

In this paper the author propose a simple model for how single spanning membrane proteins attach to lipid drafts. The authors basically propose that the affinity is related to the surface area of the sidechains and not to particular sequence features. The authors show this convincingly for a small number of experimental peptides. Although this problem has been studied somewhat before I do find the paper interesting.

We thank the reviewer for their comments. To our knowledge, there has been no systematic attempt to identify generalized, predictive determinants of raft partitioning. The model we propose does so for ~70 TMDs tested originally and several more tested for this revision, which we believe is a robust demonstration of the principle.

4-1. However, the value of the paper would significantly be increased if a more through analysis of the sequence features that effect lipid draft was analyzed. In particular I do think you could obtain equal good correlations by using average hydrophobicity as with ASA, i.e. it might not really be the surface area but hydrophobicity that is important. This should be tested.

This is a very important point. For all TMD residues we originally tested, the hydrophobicity of the TMD side chains was strongly correlated to their ASA, and thus hydrophobicity was also strongly correlated with raft affinity (black circles in Fig S4B). This is an obvious consequence of the fact that nearly all TMD residues are hydrophobic, so larger also means more hydrophobic. We attempted to break this correlation by introducing large charged or slightly hydrophilic residues into TMDs. Specifically, we added two Lys residues into the allL construct (allL2K) and 4 Trp residues into the allA8L construct (allA8L4W). In both cases, the observed raft partitioning was much closer to predictions from the ASA of the TMD rather than the hydrophobicity (as defined by the scale of Kyte and Doolittle²⁸). These results, shown below and included as Fig S4, provide strong evidence that it is indeed side chain surface area and not hydrophobicity which governs raft affinity.

Figure S4. TMD surface area rather than hydrophobicity is a determinant of raft affinity. (A) Strong correlation between TMD surface area and raft affinity established for TMDs consisting only of Ala and Leu (black circles; also shown as Fig 2C) also holds for TMDs containing charged (allL2K; blue square) or hydrophilic residues (allA8L4W; red square). (B) The raft affinity of these hydrophilic residue-containing TMDs is not well described by their aggregate hydrophobicity (calculated via the scale of Kyte and Doolittle²⁸). (C) Scatter plot showing raft affinity measurements for individual vesicles and the predictions from the correlations shown in panels A-B.

4-2. Further it would be interesting to analyze how the length of the hydrophobic region effects the results.

We have extensively analyzed the relationship between the length of the hydrophobic TMD and its raft partitioning in our previous work¹. Here, those empirical observations were rationalized with a physical model (i.e. the “mattress” model of hydrophobic mismatch; see Fig S5) and incorporated into the comprehensive model for raft affinity of TMDs (Eqn 2 in manuscript). Finally, the effect of changing TMD length on partitioning and subcellular localization was directly measured, that data is shown in Fig 3F and 4F, respectively.

- 1 Diaz-Rohrer, B. B., Levental, K. R., Simons, K. & Levental, I. Membrane raft association is a determinant of plasma membrane localization. *Proc Natl Acad Sci U S A* **111**, 8500-8505, (2014).
- 2 Pfeifer, P. Fractal dimension as working tool for surface-roughness problems *Applications of Surface Science* **18**, 146-164, (1984).
- 3 Sanner, M. F., Olson, A. J. & Spehner, J. C. Reduced surface: an efficient way to compute molecular surfaces. *Biopolymers* **38**, 305-320, (1996).
- 4 Rawicz, W. *et al.* Effect of chain length and unsaturation on elasticity of lipid bilayers. *Biophys J* **79**, 328-339, (2000).
- 5 Komura, N. *et al.* Raft-based interactions of gangliosides with a GPI-anchored receptor. *Nat Chem Biol* **12**, 402-410, (2016).
- 6 Kinoshita, M. *et al.* Raft-based sphingomyelin interactions revealed by new fluorescent sphingomyelin analogs. *J Cell Biol* **216**, 1183-1204, (2017).
- 7 Feigenson, G. W. & Buboltz, J. T. Ternary phase diagram of dipalmitoyl-PC/dilauroyl-PC/cholesterol: nanoscopic domain formation driven by cholesterol. *Biophys J* **80**, 2775-2788, (2001).
- 8 Kaiser, H. J. *et al.* Order of lipid phases in model and plasma membranes. *Proc Natl Acad Sci U S A* **106**, 16645-16650, (2009).
- 9 Zhou, Y. *et al.* Bile acids modulate signaling by functional perturbation of plasma membrane domains. *J. Biol. Chem.* **288**, 35660-35670, (2013).
- 10 Sezgin, E. *et al.* Partitioning, diffusion, and ligand binding of raft lipid analogs in model and cellular plasma membranes. *Biochim Biophys Acta* **1818**, 1777-1784, (2012).
- 11 Levental, K. R. *et al.* Polyunsaturated lipids regulate membrane domain stability by tuning membrane order. *Biophys J* **110**(8), 1800-1810, (2016).
- 12 Stone, M. B. *et al.* Protein sorting by lipid phase-like domains supports emergent signaling function in B lymphocyte plasma membranes. *eLife* **6**, (2017).
- 13 Machta, B. B. *et al.* Conditions that Stabilize Membrane Domains Also Antagonize n-Alcohol Anesthesia. *Biophys J* **111**, 537-545, (2016).
- 14 Sharpe, H. J., Stevens, T. J. & Munro, S. A comprehensive comparison of transmembrane domains reveals organelle-specific properties. *Cell* **142**, 158-169, (2010).
- 15 Lingwood, D. & Simons, K. Lipid rafts as a membrane-organizing principle. *Science* **327**, 46-50, (2010).
- 16 Levental, I. *et al.* Palmitoylation regulates raft affinity for the majority of integral raft proteins. *Proc Natl Acad Sci U S A* **107**, 22050-22054, (2010).
- 17 Machta, B. B., Papanikolaou, S., Sethna, J. P. & Veatch, S. L. Minimal model of plasma membrane heterogeneity requires coupling cortical actin to criticality. *Biophys J* **100**, 1668-1677, (2011).

- 18 Honigmann, A. *et al.* A lipid bound actin meshwork organizes liquid phase separation in model membranes. *eLife* **3**, e01671, (2014).
- 19 Arumugam, S., Petrov, E. P. & Schwille, P. Cytoskeletal pinning controls phase separation in multicomponent lipid membranes. *Biophys J* **108**, 1104-1113, (2015).
- 20 Vogel, S. K., Greiss, F., Khmelinskaia, A. & Schwille, P. Control of lipid domain organization by a biomimetic contractile actomyosin cortex. *eLife* **6**, (2017).
- 21 Almen, M. S., Nordstrom, K. J., Fredriksson, R. & Schioth, H. B. Mapping the human membrane proteome: a majority of the human membrane proteins can be classified according to function and evolutionary origin. *BMC Biol* **7**, 50, (2009).
- 22 Krogh, A., Larsson, B., von Heijne, G. & Sonnhammer, E. L. Predicting transmembrane protein topology with a hidden Markov model: application to complete genomes. *Journal of molecular biology* **305**, 567-580, (2001).
- 23 Shogomori, H. *et al.* Palmitoylation and intracellular domain interactions both contribute to raft targeting of linker for activation of T cells. *J Biol Chem* **280**, 18931-18942, (2005).
- 24 Levental, K. R. & Levental, I. Giant plasma membrane vesicles: models for understanding membrane organization. *Current topics in membranes* **75**, 25-57, (2015).
- 25 Levental, I., Grzybek, M. & Simons, K. Raft domains of variable properties and compositions in plasma membrane vesicles. *Proc Natl Acad Sci U S A* **108**, 11411-11416, (2011).
- 26 Lach, A. *et al.* Palmitoylation of MPP1 (membrane-palmitoylated protein 1)/p55 is crucial for lateral membrane organization in erythroid cells. *J Biol Chem* **287**, 18974-18984, (2012).
- 27 Levental, I. *et al.* Cholesterol-dependent phase separation in cell-derived giant plasma-membrane vesicles. *Biochem J* **424**, 163-167, (2009).
- 28 Kyte, J. & Doolittle, R. F. A simple method for displaying the hydropathic character of a protein. *Journal of molecular biology* **157**, 105-132, (1982).

REVIEWERS' COMMENTS:

Reviewer #1 (Remarks to the Author):

The authors have satisfactorily addressed all my comments.

Regarding the issue of originality and broad scope (points 1-9, 10 and 11), I cannot comment further. I have expressed my thoughts (especially regarding the originality of the new findings compared the studies of Chum et al. and Sharpe et al.) and the authors brought up interesting points.

Reviewer #3 (Remarks to the Author):

In the previous review to the work I stated that "The work by Leventhal and col. tackles in a systematic manner and with unprecedented detail the structural features that account for single transmembrane domain (TMD) proteins preference for lipid raft domains in mammalian cellular membranes, an important subject in biomembrane research with vast implications since the understanding the behavior of a large variety of proteins in mammalian can benefit a wide community in the life sciences. "

The Authors answered to all the concerns raised (by this as well as the other Reviewers), either by justifying the concerns raised, by changing/reorganizing/correcting the text, or by adding several new experimental data. The present version of the manuscript is significantly improved, and the conclusions are now very strongly supported by the results presented. Since I had already recognized the relevance and timeliness of the work, my opinion is that the work is suitable for publication in its present form.

Reviewer #4 (Remarks to the Author):

The paper is acceptable